# No Regrets for Learning the Prior in Bandits

**Soumya Basu**
Google

**Branislav Kveton**
Google Research

**Manzil Zaheer**
Google Research

**Csaba Szepesvári**
DeepMind / University of Alberta

## Abstract

We propose `AdaTS`, a Thompson sampling algorithm that adapts sequentially to bandit tasks that it interacts with. The key idea in `AdaTS` is to adapt to an unknown task prior distribution by maintaining a distribution over its parameters. When solving a bandit task, that uncertainty is marginalized out and properly accounted for. `AdaTS` is a fully-Bayesian algorithm that can be implemented efficiently in several classes of bandit problems. We derive upper bounds on its Bayes regret that quantify the loss due to not knowing the task prior, and show that it is small. Our theory is supported by experiments, where `AdaTS` outperforms prior algorithms and works well even in challenging real-world problems.

## 1 Introduction

We study the problem of maximizing the total reward, or minimizing the total regret, in a sequence of stochastic bandit instances [29, 4, 31]. We consider a Bayesian version of the problem, where the bandit instances are drawn from some distribution. More specifically, the learning agent interacts with $m$ bandit instances in $m$ tasks, with one instance per task. The interaction with each task is for $n$ rounds and with $K$ arms. The reward distribution of arm $i \in [K]$ in task $s \in [m]$ is $p_i(\cdot; \theta_{s,*})$, where $\theta_{s,*}$ is a shared parameter of all arms in task $s$. When arm $i$ is pulled in task $s$, the agent receives a random reward from $p_i(\cdot; \theta_{s,*})$. The parameters $\theta_{1,*}, \ldots, \theta_{m,*}$ are drawn independently of each other from a *task prior* $P(\cdot; \mu_*)$. The task prior is parameterized by an unknown *meta-parameter* $\mu_*$, which is drawn from a *meta-prior* $Q$. The agent does not know $\mu_*$ or $\theta_{1,*}, \ldots, \theta_{m,*}$. However, it knows $Q$ and the parametric forms of all distributions, which help it to learn about $\mu_*$. This is a form of *meta-learning* [39, 40, 7, 8], where the agent learns to act from interactions with bandit instances.

A simple approach is to ignore the hierarchical structure of the problem and solve each bandit task independently with some bandit algorithm, such as *Thompson sampling (TS)* [38, 11, 2, 37]. This may be highly suboptimal. To illustrate this, imagine that arm $1$ is optimal for any $\theta$ in the support of $P(\cdot; \mu_*)$. If $\mu_*$ was known, any reasonable algorithm would only pull arm $1$ and have zero regret over any horizon. Likewise, a clever algorithm that learns $\mu_*$ should eventually pull arm $1$ most of the time, and thus have diminishing regret as it interacts with a growing number of tasks. Two challenges arise when designing the clever algorithm. First, can it be computationally efficient? Second, what is the regret due to adapting to $\mu_*$?

We make the following contributions. First, we propose a Thompson sampling algorithm for our problem, which we call `AdaTS`. `AdaTS` maintains a distribution over the meta-parameter $\mu_*$, which concentrates over time and is marginalized out when interacting with individual bandit instances. Second, we propose computationally-efficient implementations of `AdaTS` for multi-armed bandits [29, 4], linear bandits [14, 1], and combinatorial semi-bandits [19, 13, 26]. These implementations are for specific reward distributions and conjugate task priors. Third, we bound the $n$-round Bayes regret of `AdaTS` in linear bandits and semi-bandits, and multi-armed bandits as a special case. The Bayes regret is defined by taking an expectation over all random quantities, including $\mu_* \sim Q$. Our bounds show that not knowing $\mu_*$ has a minimal impact on the regret as the number of tasks grows, of only $\tilde{O}(\sqrt{mn})$. This is in a sharp contrast to prior work [28], where this is $\tilde{O}(\sqrt{mn^2})$. Finally,

35th Conference on Neural Information Processing Systems (NeurIPS 2021).

our experiments show that AdaTS quickly adapts to the unknown meta-parameter $\mu_*$, is robust to meta-prior misspecification, and performs well even in challenging classification problems.

We present a general framework for learning to explore from similar past exploration problems. One potential application is cold-start personalization in recommender systems where users are tasks. The users have similar preferences, but neither the individual preferences nor their similarity is known in advance. Another application could be online regression with bandit feedback (Appendix E.2) where individual regression problems are tasks. Similar examples in the tasks have similar mean responses, which are unknown in advance.

## 2 Setting

We first introduce our notation. The set $\{1, \dots, n\}$ is denoted by $[n]$. The indicator $\mathbb{1}\{E\}$ denotes that event $E$ occurs. The $i$-th entry of vector $v$ is $v_i$. If the vector or its index are already subindexed, we write $v(i)$. We use $\tilde{O}$ for the big-O notation up to polylogarithmic factors. A diagonal matrix with entries $v$ is denoted $\mathrm{diag}\,(v)$. We use the terms "arm" and "action" interchangeably, depending on the context.

Our setting was proposed in Kveton et al. [28] and is defined as follows. Each bandit *problem instance* has $K$ arms. Each arm $i \in [K]$ is defined by distribution $p_i(\cdot; \theta)$ with parameter $\theta \in \Theta$. The parameter $\theta$ is shared among all arms. The mean of $p_i(\cdot; \theta)$ is denoted by $r(i; \theta)$. The learning agent interacts with $m$ instances, one at each of $m$ *tasks*. At the beginning of task $s \in [m]$, an instance $\theta_{s,*} \in \Theta$ is sampled i.i.d. from a *task prior* $P(\cdot; \mu_*)$, which is parameterized by $\mu_*$. The agent interacts with $\theta_{s,*}$ for $n$ rounds. In round $t \in [n]$, it pulls one arm and observes a stochastic realization of its reward. We denote the pulled arm in round $t$ of task $s$ by $A_{s,t} \in [K]$, the realized rewards of all arms in round $t$ of task $s$ by $Y_{s,t} \in \mathbb{R}^K$,

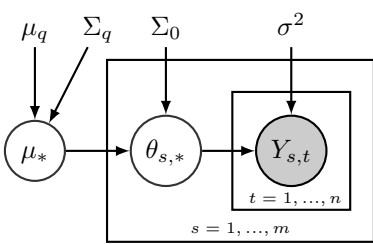

Figure 1: Graphical model of our environment.

and the reward of arm $i \in [K]$ by $Y_{s,t}(i) \sim p_i(\cdot; \theta_{s,*})$. We assume that the realized rewards $Y_{s,t}$ are i.i.d. with respect to both $s$ and $t$. A graphical model of our environment is drawn in Figure 1. We define the distribution-specific parameters $\mu_q, \Sigma_q, \Sigma_0$, and $\sigma^2$ when we instantiate our framework. Our terminology is summarized in Appendix A.

The *n-round regret* of an agent or algorithm over $m$ tasks with task prior $P(\cdot; \mu_*)$ is defined as

$$R(m, n; \mu_*) = \sum_{s=1}^{m} \mathbb{E}\left[ \sum_{t=1}^{n} r(A_{s,*}; \theta_{s,*}) - r(A_{s,t}; \theta_{s,*}) \middle| \mu_* \right], \qquad (1)$$

where $A_{s,*} = \arg\max_{i \in [K]} r(i; \theta_{s,*})$ is the *optimal arm* in the random problem instance $\theta_{s,*}$ in task $s \in [m]$. The above expectation is over problem instances $\theta_{s,*} \sim P(\cdot; \mu_*)$, their realized rewards, and also pulled arms. Note that $\mu_*$ is fixed. Russo and Van Roy [36] showed that the Bayes regret, which matches the definition in (1) in any task, of Thompson sampling in a $K$-armed bandit with $n$ rounds is $\tilde{O}(\sqrt{Kn})$. So, when TS is applied independently in each task, $R(m, n; \mu_*) = \tilde{O}(m\sqrt{Kn})$.

Our goal is to attain a comparable regret without knowing $\mu_*$. We frame this problem in a Bayesian fashion, where $\mu_* \sim Q$ before the learning agent interacts with the first task. The agent knows $Q$ and we call it a *meta-prior*. Accordingly, we consider $R(m, n) = \mathbb{E}\left[ R(m, n; \mu_*) \right]$ as a metric and call it the *Bayes regret*. Our approach is motivated by hierarchical Bayesian models [20], where the uncertainty in prior parameters, such as $\mu_*$, is represented by another distribution, such as $Q$. In these models, $Q$ is called a *hyper-prior* and $\mu_*$ is called a *hyper-parameter*. We attempt to learn $\mu_*$ from sequential interactions with instances $\theta_{s,*} \sim P(\cdot; \mu_*)$, which are also unknown. The agent can only observe their noisy realizations $Y_{s,t}$.

## 3 Algorithm

Our algorithm is presented in this section. To describe it, we need to introduce several notions of *history*, the past interactions of the agent. We denote by $H_s = (A_{s,t}, Y_{s,t}(A_{s,t}))_{t=1}^{n}$ the history in

---
**Algorithm 1** `AdaTS`: Instance-adaptive exploration in Thompson sampling.
---
1: Initialize meta-prior $Q_0 \leftarrow Q$
2: **for** $s = 1, \dots, m$ **do**
3:      Compute meta-posterior $Q_s$ (Proposition 1)
4:      Compute uncertainty-adjusted task prior $P_s$ (Proposition 1)
5:      **for** $t = 1, \dots, n$ **do**
6:          Compute posterior of $\theta$ in task $s$, $P_{s,t}(\theta) \propto \mathcal{L}_{s,t}(\theta) P_s(\theta)$
7:          Sample $\tilde{\theta}_{s,t} \sim P_{s,t}$, pull arm $A_{s,t} \leftarrow \arg\max_{i \in [K]} r(i; \tilde{\theta}_{s,t})$, and observe $Y_{s,t}(A_{s,t})$
---

task $s$ and by $H_{1:s} = H_1 \oplus \cdots \oplus H_s$ a concatenated vector of all histories in the first $s$ tasks. The history up to round $t$ in task $s$ is $H_{s,t} = (A_{s,\ell}, Y_{s,\ell}(A_{s,\ell}))_{\ell=1}^{t-1}$ and all history up to round $t$ in task $s$ is $H_{1:s,t} = H_{1:s-1} \oplus H_{s,t}$. We denote the conditional probability distribution given history $H_{1:s,t}$ by $\mathbb{P}_{s,t}(\cdot) = \mathbb{P}(\cdot \mid H_{1:s,t})$ and the corresponding conditional expectation by $\mathbb{E}_{s,t}[\cdot] = \mathbb{E}[\cdot \mid H_{1:s,t}]$.

Our algorithm is a form of Thompson sampling [38, 11, 2, 37]. TS pulls arms proportionally to being optimal with respect to the posterior. In particular, let $\mathcal{L}_{s,t}(\theta) = \prod_{\ell=1}^{t-1} p_{A_{s,\ell}}(Y_{s,\ell}(A_{s,\ell}); \theta)$ be the *likelihood of observations* in task $s$ up to round $t$. If the prior $P(\cdot; \mu_*)$ was known, the posterior of instance $\theta$ in round $t$ would be $P_{s,t}^{\mathrm{TS}}(\theta) \propto \mathcal{L}_{s,t}(\theta) P(\theta; \mu_*)$. TS would sample $\tilde{\theta}_t \sim P_{s,t}^{\mathrm{TS}}$ and pull arm $A_t = \arg\max_{i \in [K]} r(i; \tilde{\theta}_t)$.

We address the case of unknown $\mu_*$. The key idea in our method is to maintain a posterior density of $\mu_*$, which we call a *meta-posterior*. This density represents uncertainty in $\mu_*$ given history. In task $s$, we denote it by $Q_s$ and define it such that $\mathbb{P}(\mu_* \in B \mid H_{1:s-1}) = \int_{\mu \in B} Q_s(\mu) \, d\kappa_1(\mu)$ holds for any set $B$, where $\kappa_1$ is the reference measure for $\mu$. We use this more general notation, as opposing to $d\mu$, because $\mu$ can be both continuous and discrete. When solving task $s$, $Q_s$ is used to compute an *uncertainty-adjusted task prior* $P_s$, which is a posterior density of $\theta_{s,*}$ given history. Formally, $P_s$ is a density such that $\mathbb{P}(\theta_{s,*} \in B \mid H_{1:s-1}) = \int_{\theta \in B} P_s(\theta) \, d\kappa_2(\theta)$ holds for any set $B$, where $\kappa_2$ is the reference measure for $\theta$. After computing $P_s$, we run TS with prior $P_s$ to solve task $s$. To maintain $Q_s$ and $P_s$, we find it useful expressing them using a recursive update rule below.

**Proposition 1.** *Let* $\mathcal{L}_s(\theta) = \prod_{\ell=1}^n p_{A_{s,\ell}}(Y_{s,\ell}(A_{s,\ell}); \theta)$ *be the likelihood of observations in task* $s$. *Then for any task* $s \in [m]$,

$$P_s(\theta) = \int_\mu P(\theta; \mu) Q_s(\mu) \, d\kappa_1(\mu), \quad Q_s(\mu) = \int_\theta \mathcal{L}_{s-1}(\theta) P(\theta; \mu) \, d\kappa_2(\theta) \, Q_{s-1}(\mu).$$

The claim is proved in Appendix A. The proof uses the Bayes rule, where we carefully account for the fact that the observations are collected adaptively, the pulled arm in round $t$ of task $s$ depends on history $H_{1:s,t}$. The pseudocode of our algorithm is in Algorithm 1. Since the algorithm adapts to the unknown task prior $P(\cdot; \mu_*)$, we call it `AdaTS`. `AdaTS` can be implemented efficiently when $P_s$ is a conjugate prior for rewards, or a mixture of conjugate priors. We discuss several exact and efficient implementations starting from Section 3.1.

The design of `AdaTS` is motivated by `MetaTS` [28], which also maintains a meta-posterior $Q_s$. The difference is that `MetaTS` samples $\tilde{\mu}_s \sim Q_s$ in task $s$ to be optimistic with respect to the unknown $\mu_*$. Then it runs TS with prior $P(\cdot; \tilde{\mu}_s)$. While simple and intuitive, the sampling of $\tilde{\mu}_s$ induces a high variance and leads to a conservative worst-case analysis. We improve `MetaTS` by avoiding the sampling step. This leads to tighter and more general regret bounds (Section 4), beyond multi-armed bandits; while the practical performance also improves significantly (Section 5).

## 3.1 Gaussian Bandit

We start with a $K$-armed Gaussian bandit with mean arm rewards $\theta \in \mathbb{R}^K$. The reward distribution of arm $i$ is $p_i(\cdot; \theta) = \mathcal{N}(\cdot; \theta_i, \sigma^2)$, where $\sigma > 0$ is reward noise and $\theta_i$ is the mean reward of arm $i$. A natural conjugate prior for this problem class is $P(\cdot; \mu) = \mathcal{N}(\cdot; \mu, \Sigma_0)$, where $\Sigma_0 = \mathrm{diag}\left((\sigma_{0,i}^2)_{i=1}^K\right)$ is known and we learn $\mu \in \mathbb{R}^K$.

Because the prior is a multivariate Gaussian, `AdaTS` can be implemented efficiently with a Gaussian meta-prior $Q(\cdot) = \mathcal{N}(\cdot; \mu_q, \Sigma_q)$, where $\mu_q = (\mu_{q,i})_{i=1}^K$ and $\Sigma_q = \mathrm{diag}\left((\sigma_{q,i}^2)_{i=1}^K\right)$ are known mean

parameter vector and covariance matrix, respectively. In this case, the meta-posterior in task $s$ is also a Gaussian $Q_s(\cdot) = \mathcal{N}(\cdot; \hat{\mu}_s, \hat{\Sigma}_s)$, where $\hat{\mu}_s = (\hat{\mu}_{s,i})_{i=1}^{K}$ and $\hat{\Sigma}_s = \text{diag}\left((\hat{\sigma}_{s,i}^2)_{i=1}^{K}\right)$ are defined as

$$\hat{\mu}_{s,i} = \hat{\sigma}_{s,i}^2 \left( \frac{\mu_{q,i}}{\sigma_{q,i}^2} + \sum_{\ell=1}^{s-1} \frac{T_{\ell,i}}{T_{\ell,i}\,\sigma_{0,i}^2 + \sigma^2} \frac{B_{\ell,i}}{T_{\ell,i}} \right), \quad \hat{\sigma}_{s,i}^{-2} = \sigma_{q,i}^{-2} + \sum_{\ell=1}^{s-1} \frac{T_{\ell,i}}{T_{\ell,i}\,\sigma_{0,i}^2 + \sigma^2} . \quad (2)$$

Here $T_{\ell,i} = \sum_{t=1}^{n} \mathbb{1}\{A_{\ell,t} = i\}$ is the number of pulls of arm $i$ in task $\ell$ and the total reward from these pulls is $B_{\ell,i} = \sum_{t=1}^{n} \mathbb{1}\{A_{\ell,t} = i\}\, Y_{\ell,t}(i)$. The above formula has a very nice interpretation. The posterior mean $\hat{\mu}_{s,i}$ of the meta-parameter of arm $i$ is a weighted sum of the noisy estimates of the means of arm $i$ from the past tasks $B_{\ell,i}/T_{\ell,i}$ and the prior. In this sum, each bandit task is essentially a single observation. The weights are proportional to the number of pulls in a task, giving the task with more pulls a higher weight. They vary from $(\sigma_{0,i}^2 + \sigma^2)^{-1}$, when the arm is pulled only once, up to $\sigma_{0,i}^{-2}$. This is the minimum amount of uncertainty that cannot be reduced by more pulls.

The update in (2) is by Lemma 7 in Appendix A, which we borrow from Kveton et al. [28]. From Proposition 1, we have that the uncertainty-adjusted prior for task $s$ is $P_s(\cdot) = \mathcal{N}(\cdot; \hat{\mu}_s, \hat{\Sigma}_s + \Sigma_0)$.

## 3.2 Linear Bandit with Gaussian Rewards

Now we generalize Section 3.1 and consider a *linear bandit* [14, 1] with $K$ arms and $d$ dimensions. Let $\mathcal{A} \subset \mathbb{R}^d$ be an *action set* such that $|\mathcal{A}| = K$. We refer to each $a \in \mathcal{A}$ as an *arm*. Then, with a slight abuse of notation from Section 2, the reward distribution of arm $a$ is $p_a(\cdot; \theta) = \mathcal{N}(\cdot; a^\top \theta, \sigma^2)$, where $\theta \in \mathbb{R}^d$ is shared by all arms and $\sigma > 0$ is reward noise. A conjugate prior for this problem class is $P(\cdot; \mu) = \mathcal{N}(\cdot; \mu, \Sigma_0)$, where $\Sigma_0 \in \mathbb{R}^{d \times d}$ is known and we learn $\mu \in \mathbb{R}^d$.

As in Section 3.1, AdaTS can be implemented efficiently with a meta-prior $Q(\cdot) = \mathcal{N}(\cdot; \mu_q, \Sigma_q)$, where $\mu_q \in \mathbb{R}^d$ is a known mean parameter vector and $\Sigma_q \in \mathbb{R}^{d \times d}$ is a known covariance matrix. In this case, $Q_s(\cdot) = \mathcal{N}(\cdot; \hat{\mu}_s, \hat{\Sigma}_s)$, where

$$\hat{\mu}_s = \hat{\Sigma}_s \left( \Sigma_q^{-1} \mu_q + \sum_{\ell=1}^{s-1} \frac{B_\ell}{\sigma^2} - \frac{G_\ell}{\sigma^2} \left( \Sigma_0^{-1} + \frac{G_\ell}{\sigma^2} \right)^{-1} \frac{B_\ell}{\sigma^2} \right),$$

$$\hat{\Sigma}_s^{-1} = \Sigma_q^{-1} + \sum_{\ell=1}^{s-1} \frac{G_\ell}{\sigma^2} - \frac{G_\ell}{\sigma^2} \left( \Sigma_0^{-1} + \frac{G_\ell}{\sigma^2} \right)^{-1} \frac{G_\ell}{\sigma^2} .$$

Here $G_\ell = \sum_{t=1}^{n} A_{\ell,t} A_{\ell,t}^\top$ is the outer product of the feature vectors of the pulled arms in task $\ell$ and $B_\ell = \sum_{t=1}^{n} A_{\ell,t} Y_{\ell,t}(A_{\ell,t})$ is their sum weighted by their rewards. The above update follows from Lemma 7 in Appendix A, which is due to Kveton et al. [28]. From Proposition 1, the uncertainty-adjusted prior for task $s$ is $P_s(\cdot) = \mathcal{N}(\cdot; \hat{\mu}_s, \hat{\Sigma}_s + \Sigma_0)$.

We note in passing that when $K = d$ and $\mathcal{A}$ is the standard Euclidean basis of $\mathbb{R}^d$, the linear bandit reduces to a $K$-armed bandit. Since the covariance matrices are unrestricted here, the formulation in this section also shows how to generalize Section 3.1 to arbitrary covariance matrices.

## 3.3 Semi-Bandit with Gaussian Rewards

A *stochastic combinatorial semi-bandit* [19, 12, 25, 26, 42], or *semi-bandit* for short, is a $K$-armed bandit where at most $L \leq K$ arms are pulled in each round. After the arms are pulled, the agent observes their individual rewards and its reward is the sum of the individual rewards. Semi-bandits can be used to solve online combinatorial problems, such as learning to route.

We consider a Gaussian reward distribution for each arm, as in Section 3.1. The difference in the semi-bandit formulation is that the *action set* is $\mathcal{A} \subseteq \Pi_L(K)$, where $\Pi_L(K)$ is the set of all subsets of $[K]$ of size at most $L$. In round $t$ of task $s$, the agents pulls arms $A_{s,t} \in \mathcal{A}$. The meta-posterior is updated analogously to Section 3.1. The only difference is that $\mathbb{1}\{A_{\ell,t} = i\}$ becomes $\mathbb{1}\{i \in A_{\ell,t}\}$.

## 3.4 Exponential-Family Bandit with Mixture Priors

We consider a general $K$-armed bandit with mean arm rewards $\theta \in \mathbb{R}^K$. The reward distribution of arm $i$ is any one-dimensional exponential-family distribution parameterized by $\theta_i$. In a Bernoulli

bandit, this would be $p_i(\cdot; \theta) = \mathrm{Ber}(\cdot; \theta_i)$. A natural prior for this reward model would be a product of per-arm conjugate priors, such as the product of betas for Bernoulli rewards.

It is challenging to generalize our approach beyond Gaussian models because we require more than the standard notion of conjugacy. Specifically, to apply `AdaTS` to an exponentially-family prior, such as the product of betas, we need a computationally tractable prior for that prior. In this case, it does not exist. We circumvent this issue by discretization. More specifically, let $\{P(\cdot; j)\}_{j=1}^{L}$ be a set of $L$ potential conjugate priors, where each $P(\cdot; j)$ is a product of one-dimensional exponential-family priors. Then a suitable meta-prior is a vector of initial beliefs into each potential prior. In particular, it is $Q(\cdot) = \mathrm{Cat}(\cdot; w_q)$, where $w_q \in \Delta_{L-1}$ is the belief and $\Delta_L$ is the $L$-simplex.

In this case, $Q_s(j) = \int_\theta \mathcal{L}_{s-1}(\theta) P(\theta; j) \, d\kappa_2(\theta) \, Q_{s-1}(j)$ in Proposition 1 has a closed form, since it is a standard conjugate posterior update for a distribution over $\theta$ followed by integrating out $\theta$. In addition, $P_s(\theta) = \sum_{j=1}^{L} Q_s(j) P(\theta; j)$ is a mixture of exponential-family priors over $\theta$. This is an instance of latent bandits [22]. For these problems, Thompson sampling can be implemented exactly and efficiently. We do not analyze this setting because prior-dependent Bayes regret bounds for this problem class do not exist yet.

# 4 Regret Bounds

We first introduce common notation used in our proofs. The action set $\mathcal{A} \subseteq \mathbb{R}^d$ is fixed. Recall that a matrix $X \in \mathbb{R}^{d \times d}$ is *positive semi-definite (PSD)* if it is symmetric and its smallest eigenvalue is non-negative. For such $X$, we define $\sigma_{\max}^2(X) = \max_{a \in \mathcal{A}} a^\top X a$. Although $\sigma_{\max}^2(X)$ depends on $\mathcal{A}$, we suppress this dependence because $\mathcal{A}$ is fixed. We denote by $\lambda_1(X)$ the maximum eigenvalue of $X$ and by $\lambda_d(X)$ the minimum eigenvalue of $X$.

We also need basic quantities from information theory. For two probability measures $P$ and $Q$ over a common measurable space, we use $D(P||Q)$ to denote the relative entropy of $P$ with respect to $Q$. It is defined as $D(P||Q) = \int \log(\frac{dP}{dQ}) \, dP$, where $dP/dQ$ is the Radon-Nikodym derivative of $P$ with respect to $Q$; and is infinite when $P$ is not absolutely continuous with respect to $Q$. We slightly abuse our notation and let $P(X)$ denote the probability distribution of random variable $X$, $P(X \in \cdot)$. For jointly distributed random variables $X$ and $Y$, we let $P(X \mid Y)$ be the conditional distribution of $X$ given $Y$, $P(X \in \cdot \mid Y)$, which is $Y$-measurable and depends on random $Y$. The *mutual information* between $X$ and $Y$ is $I(X; Y) = D(P(X,Y)||P(X)P(Y))$, where $P(X)P(Y)$ is the distribution of the product of $P(X)$ and $P(Y)$. Intuitively, $I(X; Y)$ measures the amount of information that either $X$ or $Y$ provides about the other variable. For jointly distributed $X$, $Y$, and $Z$, we also need the *conditional mutual information* between $X$ and $Y$ conditioned on $Z$. We define this quantity as $I(X; Y \mid Z) = \mathbb{E}[\hat{I}(X; Y \mid Z)]$, where $\hat{I}(X; Y \mid Z) = D(P(X,Y \mid Z)||P(X \mid Z)P(Y \mid Z))$ is the *random conditional mutual information* between $X$ and $Y$ given $Z$. Note that $\hat{I}(X; Y \mid Z)$ is a function of $Z$. By the chain rule for the random conditional mutual information, $\hat{I}(X; Y_1, Y_2 \mid Z) = \mathbb{E}[\hat{I}(X; Y_1 \mid Y_2, Z) \mid Z] + \hat{I}(X; Y_2 \mid Z)$, where expectation is over $Y_2 \mid Z$. We would get the usual chain rule $I(X; Y_1, Y_2) = I(X; Y_1 \mid Y_2) + I(X; Y_2)$ without $Z$.

## 4.1 Generic Regret Bound

We start with a generic adaptation of the analysis of Lu and Van Roy [33] to our setting. In round $t$ of task $s$, we denote the pulled arm by $A_{s,t}$, its observed reward by $Y_{s,t} \sim p_{A_{s,t}}(\cdot; \theta_{s,*})$, and the suboptimality gap by $\Delta_{s,t} = r(A_{s,*}; \theta_{s,*}) - r(A_{s,t}; \theta_{s,*})$. For random variables $X$ and $Y$, we denote by $I_{s,t}(X; Y) = \hat{I}(X; Y \mid H_{1:s,t})$ the random mutual information between $X$ and $Y$ given history $H_{1:s,t}$ of all observations from the first $s-1$ tasks and the first $t-1$ rounds of task $s$. Similarly, for random variables $X$, $Y$, and $Z$, we denote by $I_{s,t}(X; Y \mid Z) = \mathbb{E}[\hat{I}(X; Y \mid Z, H_{1:s,t}) \mid H_{1:s,t}]$ the random mutual information between $X$ and $Y$ conditioned on $Z$, given history $H_{1:s,t}$. It is helpful to think of $I_{s,t}$ as the conditional mutual information of $X \mid H_{1:s,t}$, $Y \mid H_{1:s,t}$, and $Z \mid H_{1:s,t}$.

Let $\Gamma_{s,t}$ and $\epsilon_{s,t}$ be potentially history-dependent non-negative random variables such that

$$\mathbb{E}_{s,t}[\Delta_{s,t}] \le \Gamma_{s,t} \sqrt{I_{s,t}(\theta_{s,*}; A_{s,t}, Y_{s,t})} + \epsilon_{s,t} \tag{3}$$

holds almost surely. We want to keep both $\Gamma_{s,t}$ and $\epsilon_{s,t}$ "small". The following lemma provides a bound on the total regret over $n$ rounds in each of $m$ tasks in terms of $\Gamma_{s,t}$ and $\epsilon_{s,t}$.

**Lemma 2.** *Suppose that* (3) *holds for all* $s \in [m]$ *and* $t \in [n]$, *for some* $\Gamma_{s,t}, \epsilon_{s,t} \geq 0$. *In addition, let* $(\Gamma_s)_{s \in [m]}$ *and* $\Gamma$ *be non-negative constants such that* $\Gamma_{s,t} \leq \Gamma_s \leq \Gamma$ *holds for all* $s \in [m]$ *and* $t \in [n]$ *almost surely. Then*

$$R(m,n) \leq \Gamma \sqrt{mn I(\mu_*; H_{1:m})} + \sum_{s=1}^{m} \Gamma_s \sqrt{n I(\theta_{s,*}; H_s \mid \mu_*, H_{1:s-1})} + \sum_{s=1}^{m} \sum_{t=1}^{n} \mathbb{E}\left[\epsilon_{s,t}\right] .$$

The first term above is the price for learning $\mu_*$, while the second is the price for learning all $\theta_{s,*}$ when $\mu_*$ is known. Accordingly, the price for learning $\mu_*$ is negligible when the mutual information terms grow slowly with $m$ and $n$. Specifically, we show shortly in linear bandits that $\Gamma_{s,t}$ and $\epsilon_{s,t}$ can be set so that the last term of the bound is comparable to the rest, while $\Gamma_{s,t}$ grows slowly with $m$ and $n$. At the same time, $I(\mu_*; H_{1:m})$ and $I(\theta_{s,*}; H_s \mid \mu_*, H_{1:s-1})$ are only logarithmic in $m$ and $n$. Thus the price for learning $\mu_*$ is $\tilde{O}(\sqrt{mn})$ while that for learning all $\theta_{s,*}$ is $\tilde{O}(m\sqrt{n})$. Now we are ready to prove Lemma 2.

*Proof.* First, we use the chain rule of random conditional mutual information and derive

$$I_{s,t}(\theta_{s,*}; A_{s,t}, Y_{s,t}) \leq I_{s,t}(\theta_{s,*}, \mu_*; A_{s,t}, Y_{s,t}) = I_{s,t}(\mu_*; A_{s,t}, Y_{s,t}) + I_{s,t}(\theta_{s,*}; A_{s,t}, Y_{s,t} \mid \mu_*) .$$

Now we take the square root of both sides, apply $\sqrt{a+b} \leq \sqrt{a} + \sqrt{b}$ to the right-hand side, and multiply both sides by $\Gamma_{s,t}$. This yields

$$\Gamma_{s,t} \sqrt{I_{s,t}(\theta_{s,*}; A_{s,t}, Y_{s,t})} \leq \Gamma_{s,t} \sqrt{I_{s,t}(\mu_*; A_{s,t}, Y_{s,t})} + \Gamma_{s,t} \sqrt{I_{s,t}(\theta_{s,*}; A_{s,t}, Y_{s,t} \mid \mu_*)} . \quad (4)$$

We start with the second term in (4). Fix task $s$. From $\Gamma_{s,t} \leq \Gamma_s$, followed by the Cauchy-Schwarz and Jensen's inequalities, we have

$$\mathbb{E}\left[ \sum_{t=1}^{n} \Gamma_{s,t} \sqrt{I_{s,t}(\theta_{s,*}; A_{s,t}, Y_{s,t} \mid \mu_*)} \right] \leq \Gamma_s \sqrt{n \mathbb{E}\left[ \sum_{t=1}^{n} I_{s,t}(\theta_{s,*}; A_{s,t}, Y_{s,t} \mid \mu_*) \right]} .$$

Thanks to $\mathbb{E}\left[I_{s,t}(\theta_{s,*}; A_{s,t}, Y_{s,t} \mid \mu_*)\right] = I(\theta_{s,*}; A_{s,t}, Y_{s,t} \mid \mu_*, H_{1:s,t})$ and the chain rule of mutual information, we have $\mathbb{E}\left[\sum_{t=1}^{n} I_{s,t}(\theta_{s,*}; A_{s,t}, Y_{s,t} \mid \mu_*)\right] = I(\theta_{s,*}; H_s \mid \mu_*, H_{1:s-1})$.

Now we consider the first term in (4). We bound $\Gamma_{s,t}$ using $\Gamma$, then apply the Cauchy-Schwarz and Jensen's inequalities, and obtain $\mathbb{E}\left[\sum_{s=1}^{m} \sum_{t=1}^{n} \Gamma_{s,t} \sqrt{I_{s,t}(\mu_*; A_{s,t}, Y_{s,t})}\right] \leq \Gamma \sqrt{mn I(\mu_*; H_{1:m})}$; where we used the chain rule to get

$$\mathbb{E}\left[ \sum_{s=1}^{m} \sum_{t=1}^{n} I_{s,t}(\mu_*; A_{s,t}, Y_{s,t}) \right] = \sum_{s=1}^{m} \sum_{t=1}^{n} I(\mu_*; A_{s,t}, Y_{s,t} \mid H_{1:s,t}) = I(\mu_*; H_{1:m}) .$$

This completes the proof. $\qquad \square$

## 4.2 Linear Bandit with Gaussian Rewards

Now we derive regret bounds for linear bandits (Section 3.2). Without loss of generality, we make an assumption that the action set is bounded.

**Assumption 1.** *The arms are vectors in a unit ball,* $\max_{a \in \mathcal{A}} \|a\|_2 \leq 1$.

Our analysis is for AdaTS with a small amount of *forced exploration* in each task. This guarantees that our estimate of $\mu_*$ improves uniformly in all directions after each task $s$. Therefore, we assume that the action set is diverse enough to explore in all directions.

**Assumption 2.** *There exist arms* $\{a_i\}_{i=1}^{d} \subseteq \mathcal{A}$ *such that* $\lambda_d(\sum_{i=1}^{d} a_i a_i^\top) \geq \eta$ *for some* $\eta > 0$.

This assumption is without loss of generality. In particular, if such a set does not exist, the action set $\mathcal{A}$ can be projected into a lower dimensional space where the assumption holds. AdaTS is modified as follows. In each task, we initially pulls the arms $\{a_i\}_{i=1}^{d}$ to explore all directions.

We start by showing that (3) holds for suitably "small" $\Gamma_{s,t}$ and $\epsilon_{s,t}$. In AdaTS, in round $t$ of task $s$, the posterior distribution of $\theta_{s,*}$ is $\mathcal{N}(\hat{\mu}_{s,t}, \hat{\Sigma}_{s,t})$, where

$$\hat{\mu}_{s,t} = \hat{\Sigma}_{s,t}\left((\Sigma_0 + \hat{\Sigma}_s)^{-1}\hat{\mu}_s + \sum_{\ell=1}^{t-1}\frac{A_{s,\ell}Y_{s,\ell}}{\sigma^2}\right), \quad \hat{\Sigma}_{s,t}^{-1} = (\Sigma_0 + \hat{\Sigma}_s)^{-1} + \sum_{\ell=1}^{t-1}\frac{A_{s,\ell}A_{s,\ell}^\top}{\sigma^2},$$

and $\hat{\mu}_s$ and $\hat{\Sigma}_s$ are defined in Section 3.2. Then, from the properties of Gaussian distributions and that AdaTS samples from the posterior, we get a bound on $\Gamma_{s,t}$ and $\epsilon_{s,t}$ as a function of a tunable parameter $\delta \in (0, 1]$.

**Lemma 3.** *For all tasks $s \in [m]$, rounds $t \in [n]$, and any $\delta \in (0, 1]$, (3) holds almost surely for*

$$\Gamma_{s,t} = 4\sqrt{\frac{\sigma_{\max}^2(\hat{\Sigma}_{s,t})}{\log(1 + \sigma_{\max}^2(\hat{\Sigma}_{s,t})/\sigma^2)}\log(4|\mathcal{A}|/\delta)}, \quad \epsilon_{s,t} = \sqrt{2\delta\sigma_{\max}^2(\hat{\Sigma}_{s,t})} + 2\mathcal{E}_{s,t}\mathbb{E}_{s,t}[\|\theta_{s,*}\|_2],$$

*where $\mathcal{E}_{s,t}$ is the indicator of forced exploration in round $t$ of task $s$. Moreover, for each task $s$, the following history-independent bound holds almost surely,*

$$\sigma_{\max}^2(\hat{\Sigma}_{s,t}) \le \lambda_1(\Sigma_0)\left(1 + \frac{\lambda_1(\Sigma_q)\left(1 + \frac{\sigma^2}{\eta\lambda_1(\Sigma_0)}\right)}{\lambda_1(\Sigma_0) + \sigma^2/\eta + s\lambda_1(\Sigma_q)}\right). \tag{5}$$

Lemma 3 is proved in Appendix C.3. By using the bound in (5), we get that $\Gamma_{s,t} = O(\sqrt{\log(1/\delta)})$ and $\epsilon_{s,t} = O(\sqrt{\delta})$. Lemma 3 differs from Lu and Van Roy [33] in two aspects. First, it considers uncertainty in the estimate of $\mu_*$ along with $\theta_{s,*}$. Second, it does not require that the rewards are bounded. Our next lemma bounds the mutual information terms in Lemma 2, by exploiting the hierarchical structure of our linear bandit model (Figure 1).

**Lemma 4.** *For any $H_{1:s,t}$-adapted action sequence and any $s \in [m]$, we have*

$$I(\theta_{s,*}; H_s \mid \mu_*, H_{1:s-1}) \le \tfrac{d}{2}\log\left(1 + \tfrac{\lambda_1(\Sigma_0)n}{\sigma^2}\right), \quad I(\mu_*; H_{1:m}) \le \tfrac{d}{2}\log\left(1 + \tfrac{\lambda_1(\Sigma_q)m}{\lambda_d(\Sigma_0) + \sigma^2/n}\right).$$

Now we are ready to prove our regret bound for the linear bandit. We take the mutual-information bounds from Lemma 4, and the bounds on $\Gamma_{s,t}$ and $\epsilon_{s,t}$ from Lemma 3, and plug them into Lemma 2. Specifically, $\sigma_{\max}^2(\hat{\Sigma}_{s,t}) \le \lambda_1(\Sigma_q) + \lambda_1(\Sigma_0)$ holds for any $s$ and $t$ by Lemma 3, which yields $\Gamma$ in Lemma 2. On the other hand, $\Gamma_s$ is bounded using the upper bound in (5), which relies on forced exploration. Our regret bound is stated below. The terms $c_1$ to $c_4$ are at most polylogarithmic in $d$, $m$, and $n$; and thus small. The term $c_2$ arises due to summing up $\Gamma_s$ over all tasks $s$.

**Theorem 5 (Linear bandit).** *The regret of AdaTS is bounded for any $\delta \in (0, 1]$ as*

$$R(m, n) \le \underbrace{c_1\sqrt{dmn}}_{\text{Learning of } \mu_*} + (m + c_2)\underbrace{R_\delta(n; \mu_*)}_{\text{Per-task regret}} + \underbrace{c_3dm}_{\text{Forced exploration}},$$

*where*

$$c_1 = \sqrt{8\frac{\lambda_1(\Sigma_q) + \lambda_1(\Sigma_0)}{\log\left(1 + \frac{\lambda_1(\Sigma_q) + \lambda_1(\Sigma_0)}{\sigma^2}\right)}\log(4|\mathcal{A}|/\delta)\log\left(1 + \frac{\lambda_1(\Sigma_q)m}{\lambda_d(\Sigma_0) + \sigma^2/n}\right)},$$

$c_2 = \left(1 + \frac{\sigma^2}{\eta\lambda_1(\Sigma_0)}\right)\log m$, *and $c_3 = 2\sqrt{\|\mu_q\|_2^2 + \text{tr}(\Sigma_q + \Sigma_0)}$. The* per-task regret *is bounded as* $R_\delta(n; \mu_*) \le c_4\sqrt{dn} + \sqrt{2\delta\lambda_1(\Sigma_0)}n$, *where*

$$c_4 = \sqrt{8\frac{\lambda_1(\Sigma_0)}{\log\left(1 + \frac{\lambda_1(\Sigma_0)}{\sigma^2}\right)}\log(4|\mathcal{A}|/\delta)\log\left(1 + \frac{\lambda_1(\Sigma_0)n}{\sigma^2}\right)}.$$

The bound in Theorem 5 is sublinear in $n$ for $\delta = 1/n^2$. It has three terms. The first term is the regret due to learning $\mu_*$ over all tasks; and it is $\tilde{O}(\sqrt{dmn})$. The second term is the regret for acting in $m$ tasks under the assumption that $\mu_*$ is known; and it is $\tilde{O}(m\sqrt{dn})$. The last term is the regret for

forced exploration; and it is $\tilde{O}(dm)$. Overall, the extra regret due to unknown $\mu_*$ is $\tilde{O}(\sqrt{dmn} + dm)$ and is much lower than $\tilde{O}(m\sqrt{dn})$ when $d \ll n$. Therefore, we call AdaTS a *no-regret algorithm* for linear bandits. Our bound also reflects the fact that the regret decreases as both priors become more informative, $\lambda_1(\Sigma_0) \to 0$ and $\lambda_1(\Sigma_q) \to 0$.

A frequentist regret bound for linear TS with finitely-many arms is $\tilde{O}(d\sqrt{n})$ [3]. When applied to $m$ tasks, it would be $\tilde{O}(dm\sqrt{n})$ and is worse by a factor of $\sqrt{d}$ than our regret bound. To show that our bound reflects the structure of our problem, we compare AdaTS to two variants of linear TS that are applied independently to each task. The first variant knows $\mu_*$ and thus has more information. Its regret can bounded by setting $c_1 = c_2 = c_3 = 0$ in Theorem 5 and is lower than that of AdaTS. The second variant knows that $\mu_* \sim \mathcal{N}(\mu_q, \Sigma_q)$ but does not model that the tasks share $\mu_*$. This is analogous to assuming that $\theta_{s,*} \sim \mathcal{N}(\mu_q, \Sigma_q + \Sigma_0)$. The regret of this approach can be bounded by setting $c_1 = c_2 = c_3 = 0$ in Theorem 5 and replacing $\lambda_1(\Sigma_0)$ in $c_4$ by $\lambda_1(\Sigma_q + \Sigma_0)$. Since the task regret increases linearly with $m$ and $\lambda_1(\Sigma_q + \Sigma_0) > \lambda_1(\Sigma_0)$, this approach would ultimately have a higher regret than AdaTS as the number of tasks $m$ increases.

### 4.3 Semi-Bandit with Gaussian Rewards

In semi-bandits (Section 3.3), we use the independence of arms to decompose the per-round regret differently. Similarly to Section 4.2, we analyze AdaTS with forced exploration, where each arm is initially pulled at least once. This is always possible in at most $K$ rounds, since there exists at least one $a \in \mathcal{A}$ that contains any given arm.

Let $\Gamma_{s,t}(k)$ and $\epsilon_{s,t}(k)$ be non-negative history-dependent constants, for each arm $k \in [K]$, were we use $(k)$ to refer to arm-specific quantities. Then an analogous bound to (3) is

$$\mathbb{E}_{s,t}[\Delta_{s,t}] \leq \sum_{k \in [K]} \mathbb{P}_{s,t}(k \in A_{s,t}) \left( \Gamma_{s,t}(k) \sqrt{I_{s,t}(\theta_{s,*}(k); k, Y_{s,t}(k))} + \epsilon_{s,t}(k) \right).$$

The term $(k, Y_{s,t}(k))$ is a tuple of a pulled arm $k$ and its observation in round $t$ of task $s$. For any $k$, from the chain rule of mutual information, we have

$$I_{s,t}(\theta_{s,*}(k); k, Y_{s,t}(k)) \leq I_{s,t}(\mu_*(k); k, Y_{s,t}(k)) + I_{s,t}(\theta_{s,*}(k); k, Y_{s,t}(k) \mid \mu_*(k)).$$

Next we combine the mutual-information terms across all rounds and tasks, as in Lemma 2, and bound corresponding $\Gamma_{s,t}(k)$ and $\epsilon_{s,t}(k)$ independently of $m$ and $n$. Due to forced exploration, the estimate of $\mu_*(k)$ improves for all arms $k$ as more tasks are completed, and $\Gamma_{s,t}(k)$ decreases with $s$. This leads to Theorem 6, which is proved in Appendix D.

**Theorem 6 (Semi-bandit).** *The regret of AdaTS is bounded for any $\delta \in (0, 1]$ as*

$$R(m, n) \leq \underbrace{c_1 \sqrt{KLmn}}_{\text{Learning of } \mu_*} + (m + c_2) \underbrace{R_\delta(n; \mu_*)}_{\text{Per-task regret}} + \underbrace{c_3 K^{3/2} m}_{\text{Forced exploration}} + c_4 \sigma \sqrt{2\delta mn},$$

*where*

$$c_1 = 4 \sqrt{\frac{1}{K} \sum_{k \in [K]} \frac{\sigma_{q,k}^2 + \sigma_{0,k}^2}{\log\left(1 + \frac{\sigma_{q,k}^2 + \sigma_{0,k}^2}{\sigma^2}\right)} \log(4K/\delta) \log\left(1 + \frac{\sigma_{q,k}^2 m}{\sigma_{0,k}^2 + \sigma^2/n}\right)},$$

$$c_2 = \left(1 + \max_{k \in [K]: \sigma_{0,k} > 0} \frac{\sigma^2}{\sigma_{0,k}^2}\right) \log m, \quad c_3 = 2 \sqrt{\sum_{k \in [K]} (\mu_{q,k}^2 + \sigma_{q,k}^2 + \sigma_{0,k}^2)},$$

$$c_4 = \sqrt{\frac{1}{K} \sum_{k \in [K]: \sigma_{0,k} = 0} \log\left(1 + \frac{\sigma_{q,k}^2 m}{\sigma^2}\right)}.$$

*The per-task regret is bounded as $R_\delta(n; \mu_*) \leq c_5 \sqrt{KLn} + \sqrt{2\delta \frac{1}{K} \sum_{k \in [K]} \sigma_{0,k}^2} n$, where*

$$c_5 = 4 \sqrt{\frac{1}{K} \sum_{k \in [K]: \sigma_{0,k} > 0} \frac{\sigma_{0,k}^2}{\log\left(1 + \frac{\sigma_{0,k}^2}{\sigma^2}\right)} \log(4K/\delta) \log\left(1 + \frac{\sigma_{0,k}^2 n}{\sigma^2}\right)}.$$

*The prior widths $\sigma_{q,k}$ and $\sigma_{0,k}$ are defined as in Section 3.1.*

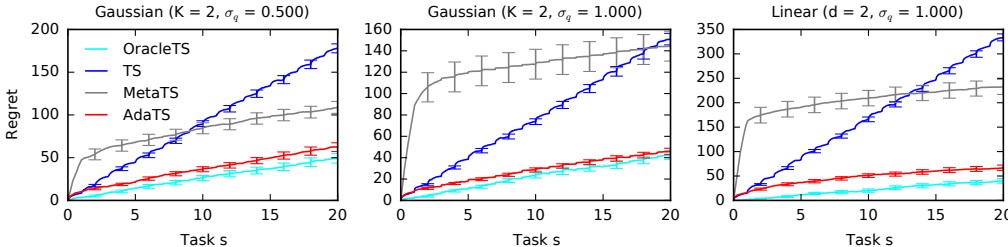

Figure 2: Comparison of `AdaTS` to three baselines on three bandit problems.

The bound in Theorem 6 is sublinear in $n$ for $\delta = 1/n^2$. Its form resembles Theorem 5. Specifically, the regret for learning $\mu_*$ is $\tilde{O}(\sqrt{KLmn})$ and for forced exploration is $\tilde{O}(K^{3/2}m)$. Both of these are much lower than the regret for learning to act in $m$ tasks when $\mu_*$ is known, $\tilde{O}(m\sqrt{KLn})$, for $K \ll Ln$. Therefore, `AdaTS` is also a *no-regret algorithm* for semi-bandits.

Theorem 6 improves upon a naive application of Theorem 5 to semi-bandits. This is because all prior width constants are averages, as opposing to the maximum over arms in Theorem 5. To the best of our knowledge, such per-arm prior dependence has not been captured in semi-bandits by any prior work. To illustrate the difference, consider a problem where $\sigma_{0,k} > 0$ for only $K' \ll K$ arms. This means that only $K'$ arms are uncertain in the tasks. Then the bound in Theorem 6 is $\tilde{O}(m\sqrt{K'Ln})$, while the bound in Theorem 5 would be $\tilde{O}(m\sqrt{KLn})$. For the arms $k$ where $\sigma_{0,k} = 0$, the regret over all tasks is sublinear in $m$.

## 5 Experiments

We experiment with two synthetic problems. In both problems, the number of tasks is $m = 20$ and each task has $n = 200$ rounds. The first problem is a Gaussian bandit (Section 3.1) with $K = 2$ arms. The meta-prior is $\mathcal{N}(\mathbf{0}, \Sigma_q)$ with $\Sigma_q = \sigma_q^2 I_K$, the prior covariance is $\Sigma_0 = \sigma_0^2 I_K$, and the reward noise is $\sigma = 1$. We experiment with $\sigma_q \geq 0.5$ and $\sigma_0 = 0.1$. Since $\sigma_q \gg \sigma_0$, the entries of $\theta_{s,*}$ are likely to have the same order as in $\mu_*$. Therefore, a clever algorithm that learns $\mu_*$ could have very low regret. The second problem is a linear bandit (Section 3.2) in $d = 2$ dimensions with $K = 5d$ arms. The action set is sampled from a unit sphere. The meta-prior, prior, and noise are the same as in the Gaussian bandit. All results are averaged over 100 runs.

`AdaTS` is compared to three baselines. The first is idealized TS with the true prior $\mathcal{N}(\mu_*, \Sigma_0)$ and we call it `OracleTS`. `OracleTS` shows the minimum attainable regret. The second is agnostic TS, which ignores the structure of the problem. We call it `TS` and implement it with prior $\mathcal{N}(\mathbf{0}, \Sigma_q + \Sigma_0)$, since $\theta_{s,*}$ can be viewed as a sample from this prior when the structure is ignored (Section 4.2). The third baseline is `MetaTS` of Kveton et al. [28]. All methods are evaluated by their cumulative regret up to task $s$, which we plot as it accumulates round-by-round within each task (Figure 2). The regret of the algorithms that do not learn $\mu_*$ (`OracleTS` and `TS`) is obviously linear in $s$, as they solve $s$ similar tasks with the same policy (Section 2). A lower slope indicates a better policy. As no algorithm can outperform `OracleTS`, no regret can grow sublinearly in $s$.

Our results are reported in Figure 2. We start with a Gaussian bandit with $\sigma_q = 0.5$. This setting is identical to Figure 1b of Kveton et al. [28]. We observe that `AdaTS` outperforms `TS`, which does not learn $\mu_*$, and is comparable to `OracleTS`, which knows $\mu_*$. Its regret is about 30% lower than that of `MetaTS`. Now we increase the meta-prior width to $\sigma_q = 1$. In this setting, meta-parameter sampling in `MetaTS` leads to high biases in earlier tasks. This leads to a major increase in regret, while `AdaTS` performs comparably to `OracleTS`. We end with a linear bandit with $\sigma_q = 1$. In this experiment, `AdaTS` outperforms `MetaTS` again and has more than three times lower regret.

Appendix E contains more experiments. In Appendix E.1, we experiment with more values of $K$ and $d$, and show the robustness of `AdaTS` to missspecified meta-prior $Q$. In Appendix E.2, we apply `AdaTS` to

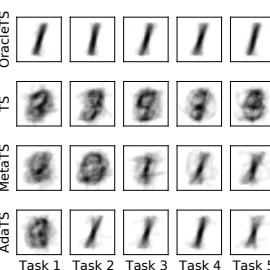

Figure 3: Meta-learning of a highly rewarding digit 1.

bandit classification problems. In Figure 3, we show results for one of these problems, meta-learning a highly rewarding digit 1 in the bandit setting. For each method and task $s$, we show the average digit corresponding to the pulled arms in round 1 of task $s$. AdaTS learns a good meta-parameter $\mu_*$ almost instantly, since its average digit in task 2 already resembles digit 1.

## 6 Related Work

Two closest related works are Bastani et al. [6] and Kveton et al. [28]. Bastani et al. [6] proposed Thompson sampling that learns the prior from a sequence of pricing experiments. The algorithm is tailored to pricing and learns through forced exploration using conservative TS. Therefore, it is conservative. Bastani et al. [6] also did not derive prior-dependent bounds.

Our studied setting is identical to Kveton et al. [28]. However, the design of AdaTS is very different from MetaTS. MetaTS samples the meta-parameter $\mu_s$ at the beginning of each task $s$ and uses it to compute the posterior of the task parameter $\theta_{s,*}$. Since $\mu_s$ is fixed within the task, MetaTS does not have a correct posterior of $\theta_{s,*}$ given the history. AdaTS marginalizes out the uncertainty in the meta-parameter $\mu_*$ and thus has a correct posterior of $\theta_{s,*}$ within the task. This seemingly minor difference leads to an approach that is more principled, comparably general, has a fully-Bayesian analysis beyond multi-armed bandits, and may have several-fold lower regret in practice. While it is possible that the analysis of MetaTS could be extended to linear bandits, the price for meta-learning would likely remain $\tilde{O}(\sqrt{m}n^2)$. This cost arises due sampling the meta-parameter $\mu_s$ at the beginning of each task $s$. The price of meta-learning in our work is mere $\tilde{O}(\sqrt{mn})$, a huge improvement.

AdaTS is a meta-learning algorithm [39, 40, 7, 8, 17, 18]. Meta-learning has a long history in multi-armed bandits. Some of the first works are Azar et al. [5] and Gentile et al. [21], who proposed UCB algorithms for multi-task learning. Deshmukh et al. [15] studied multi-task learning in contextual bandits. Cella et al. [10] proposed a UCB algorithm that meta-learns the mean parameter vector in a linear bandit, which is akin to learning $\mu_*$ in Section 3.2. Another recent work is Yang et al. [43], who studied regret minimization with multiple parallel bandit instances, with the goal of learning their shared subspace. All of these works are frequentist, analyze a stronger notion of regret, and often lead to conservative algorithm designs. In contrast, we leverage the fundamentals of Bayesian reasoning to design a general-purpose algorithm that performs well when run as analyzed.

Several recent papers approached the problem of learning a bandit algorithm using policy gradients [16, 9, 27, 44, 35], including learning Thompson sampling [27, 35]. These works focus on offline optimization against a known bandit-instance distribution and have no convergence guarantees in general [9, 27]. Tuning of bandit algorithms is known to reduce regret [41, 34, 24, 23]. Typically it is ad-hoc and we believe that meta-learning is a proper way of framing this problem.

## 7 Conclusions

We propose AdaTS, a fully-Bayesian algorithm for meta-learning in bandits that adapts to a sequence of bandit tasks that it interacts with. AdaTS attains low regret by adapting the uncertainty in both the meta and per-task parameters. We analyze the Bayes regret of AdaTS using information-theory tools that isolate the effect of learning the meta-parameter from that of learning the per-task parameters. For linear bandits and semi-bandits, we derive novel prior-dependent regret bounds that show that the price for learning the meta-parameter is low. Our experiments underscore the generality of AdaTS, good out-of-the-box performance, and robustness to meta-prior misspecification.

We leave open several questions of interest. For instance, except for Section 3.4, our algorithms are for Gaussian rewards and priors, and so are their regret analyses. An extension beyond Gaussians would be of both practical and theoretical value. Our current work also relies heavily on a particular parameterization of tasks, where the mean $\theta_{s,*}$ is unknown but the covariance $\Sigma_0$ is known. It is not immediately obvious if a computationally-efficient extension to unknown $\Sigma_0$ exists.

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
