# A Algorithm Details

Our terminology is summarized below:

| | |
|---|---|
| $\theta_{s,*}$ | Bandit instance parameter in task $s$, generated as $\theta_{s,*} \sim P(\cdot; \mu_*)$ |
| $P(\cdot; \mu_*)$ | Task prior, a distribution over bandit instance parameter $\theta_{s,*}$ |
| $\mu_*$ | Meta-parameter, a parameter of the task distribution |
| $Q$ | Meta-prior, a distribution over the meta-parameter $\mu_*$ |
| $P_s$ | Uncertainty-adjusted prior in task $s$, a distribution over $\theta_{s,*}$ conditioned on $H_{1:s-1}$ |
| $Q_s$ | Meta-posterior in task $s$, a distribution over $\mu_*$ conditioned on $H_{1:s-1}$ |
| $Y_{s,t}$ | Stochastic rewards of all arms in round $t$ of task $s$ |
| $A_{s,t}$ | Pulled arm in round $t$ of task $s$ |

We continue with two lemmas, which are used in the algorithmic part of the paper (Section 3).

**Proposition 1.** *Let $\mathcal{L}_s(\theta) = \prod_{\ell=1}^{n} p_{A_{s,\ell}}(Y_{s,\ell}(A_{s,\ell}); \theta)$ be the likelihood of observations in task $s$. Then for any task $s \in [m]$,*

$$P_s(\theta) = \int_\mu P(\theta; \mu) \, Q_s(\mu) \, d\kappa_1(\mu), \quad Q_s(\mu) = \int_\theta \mathcal{L}_{s-1}(\theta) \, P(\theta; \mu) \, d\kappa_2(\theta) \, Q_{s-1}(\mu).$$

*Proof.* To simplify presentation, our proof is under the assumption that $\theta_{s,*}$ and $\mu_*$ take on countably-many values. A more general measure-theory treatment, where we would maintain measures over $\theta_{s,*}$ and $\mu_*$, would follow the same line of reasoning; and essentially replace all probabilities with densities. A good discussion of this topic is in Section 34 of Lattimore and Szepesvari [31].

The following convention is used in the proof. The values of random variables that we marginalize out, such as $\theta_{s,*}$ and $\mu_*$, are explicitly assigned. For fixed variables, such as the history $H_{1:s-1}$, we also treat $H_{1:s-1}$ as the actual value assigned to $H_{1:s-1}$.

We start with the posterior distribution of $\theta_{s,*}$ in task $s$, which can be expressed as

$$\mathbb{P}(\theta_{s,*} = \theta \mid H_{1:s-1}) = \sum_\mu \mathbb{P}(\theta_{s,*} = \theta, \mu_* = \mu \mid H_{1:s-1})$$

$$= \sum_\mu \mathbb{P}(\theta_{s,*} = \theta \mid \mu_* = \mu) \, \mathbb{P}(\mu_* = \mu \mid H_{1:s-1}).$$

The second equality holds because $\theta_{s,*}$ is independent of history $H_{1:s-1}$ given $\mu_*$. Now note that $\mathbb{P}(\mu_* = \mu \mid H_{1:s-1})$ is the meta-posterior in task $s$. It can be rewritten as

$$\mathbb{P}(\mu_* = \mu \mid H_{1:s-1}) = \frac{\mathbb{P}(\mu_* = \mu \mid H_{1:s-1})}{\mathbb{P}(\mu_* = \mu \mid H_{1:s-2})} \mathbb{P}(\mu_* = \mu \mid H_{1:s-2})$$

$$= \frac{\mathbb{P}(H_{s-1} \mid H_{1:s-2}, \mu_* = \mu)}{\mathbb{P}(H_{s-1} \mid H_{1:s-2})} \mathbb{P}(\mu_* = \mu \mid H_{1:s-2})$$

$$\propto \underbrace{\mathbb{P}(H_{s-1} \mid H_{1:s-2}, \mu_* = \mu)}_{f_1(\mu)} \mathbb{P}(\mu_* = \mu \mid H_{1:s-2}),$$

where $\mathbb{P}(\mu_* = \mu \mid H_{1:s-2})$ is the meta-posterior in task $s-1$. The last step follows from the fact that $\mathbb{P}(H_{s-1} \mid H_{1:s-2})$ is constant in $\mu$. Now we focus on $f_1(\mu)$ above and rewrite it as

$$f_1(\mu) = \sum_\theta \mathbb{P}(H_{s-1}, \theta_{s-1,*} = \theta \mid H_{1:s-2}, \mu_* = \mu)$$

$$= \sum_\theta \mathbb{P}(H_{s-1} \mid H_{1:s-2}, \theta_{s-1,*} = \theta, \mu_* = \mu) \, \mathbb{P}(\theta_{s-1,*} = \theta \mid H_{1:s-2}, \mu_* = \mu)$$

$$= \sum_\theta \underbrace{\mathbb{P}(H_{s-1} \mid H_{1:s-2}, \theta_{s-1,*} = \theta)}_{f_2(\theta)} \mathbb{P}(\theta_{s-1,*} = \theta \mid \mu_* = \mu).$$

In the last step, we use that the history $H_{s-1}$ is independent of $\mu_*$ given $H_{1:s-2}$ and $\theta_{s-1,*}$, and that the task parameter $\theta_{s-1,*}$ is independent of $H_{1:s-2}$ given $\mu_*$.

Now we focus on $f_2(\theta)$ above. To simplify notation, it is useful to define $Y_t = Y_{s-1,t}(A_{s-1,t})$ and $A_t = A_{s-1,t}$. Then we can rewrite $f_2(\theta)$ as

$$f_2(\theta) = \prod_{t=1}^{n} \mathbb{P}\left(A_t, Y_t \mid H_{1:s-1,t}, \theta_{s-1,*} = \theta\right)$$

$$= \prod_{t=1}^{n} \mathbb{P}\left(Y_t \mid A_t, H_{1:s-1,t}, \theta_{s-1,*} = \theta\right) \mathbb{P}\left(A_t \mid H_{1:s-1,t}, \theta_{s-1,*} = \theta\right)$$

$$= \prod_{t=1}^{n} \mathbb{P}\left(Y_t \mid A_t, \theta_{s-1,*} = \theta\right) \mathbb{P}\left(A_t \mid H_{1:s-1,t}\right) \propto \mathbb{P}\left(Y_{1:n} \mid A_{1:n}, \theta_{s-1,*} = \theta\right) .$$

In the third equality, we use that the reward $Y_t$ is independent of history $H_{1:s-1,t}$ given the pulled arm $A_t$ and task parameter $\theta_{s-1,*}$, and that $A_t$ is independent of $\theta_{s-1,*}$ given $H_{1:s-1,t}$. In the last step, we use that $\mathbb{P}\left(A_t \mid H_{1:s-1,t}\right)$ is constant in $\theta$.

Finally, we combine all above claims, note that

$$\mathbb{P}\left(\theta_{s,*} = \theta \mid \mu_* = \mu\right) = \mathbb{P}\left(\theta_{s-1,*} = \theta \mid \mu_* = \mu\right) = P(\theta; \mu),$$

and get

$$\mathbb{P}\left(\theta_{s,*} = \theta \mid H_{1:s-1}\right) = \sum_{\mu} P(\theta; \mu) \mathbb{P}\left(\mu_* = \mu \mid H_{1:s-1}\right) ,$$

$$\mathbb{P}\left(\mu_* = \mu \mid H_{1:s-1}\right) = \sum_{\theta} \mathbb{P}\left(Y_{1:n} \mid A_{1:n}, \theta_{s-1,*} = \theta\right) P(\theta; \mu) \mathbb{P}\left(\mu_* = \mu \mid H_{1:s-2}\right) .$$

These are the claims that we wanted to prove, since

$$P_s(\theta) = \mathbb{P}\left(\theta_{s,*} = \theta \mid H_{1:s-1}\right) ,$$
$$Q_s(\mu) = \mathbb{P}\left(\mu_* = \mu \mid H_{1:s-1}\right) ,$$
$$\mathcal{L}_{s-1}(\theta) = \mathbb{P}\left(Y_{1:n} \mid A_{1:n}, \theta_{s-1,*} = \theta\right) .$$

This concludes the proof. $\qquad \square$

**Lemma 7.** *Fix integers $s$ and $n$, features $(x_{\ell,t})_{\ell \in [s], t \in [n]}$, and consider a generative process*

$$\mu_* \sim \mathcal{N}(\mu_q, \Sigma_q) ,$$
$$\forall \ell \in [s] : \theta_{\ell,*} \mid \mu_* \sim \mathcal{N}(\mu_*, \Sigma_0) ,$$
$$\forall \ell \in [s], t \in [n] : Y_{\ell,t} \mid \mu_* \sim \mathcal{N}(x_{\ell,t}^\top \theta_{\ell,*}, \sigma^2) ,$$

*where all variables are drawn independently. Then $\mu_* \mid (Y_{\ell,t})_{\ell \in [s], t \in [n]} \sim \mathcal{N}(\hat{\mu}, \hat{\Sigma})$ for*

$$\hat{\mu} = \hat{\Sigma} \left( \Sigma_q^{-1} \mu_q + \sum_{\ell=1}^{s} \frac{B_\ell}{\sigma^2} - \frac{G_\ell}{\sigma^2} \left( \Sigma_0^{-1} + \frac{G_\ell}{\sigma^2} \right)^{-1} \frac{B_\ell}{\sigma^2} \right) ,$$

$$\hat{\Sigma}^{-1} = \Sigma_q^{-1} + \sum_{\ell=1}^{s} \frac{G_\ell}{\sigma^2} - \frac{G_\ell}{\sigma^2} \left( \Sigma_0^{-1} + \frac{G_\ell}{\sigma^2} \right)^{-1} \frac{G_\ell}{\sigma^2} ,$$

*where $G_\ell = \sum_{t=1}^{n} x_{\ell,t} x_{\ell,t}^\top$ is the outer product of the features in task $\ell$ and $B_\ell = \sum_{t=1}^{n} x_{\ell,t} Y_{\ell,t}$ is their sum weighted by observations.*

*Proof.* The claim is proved in Appendix D of Kveton et al. [28]. We restate it for completeness. $\quad \square$

# B Proofs for Section 4.1: Generic Regret Bound

## B.1 Preliminaries and Omitted Definitions

**Notation for History:** Let us recall that $H_{s,t} = ((A_{s,1}, Y_{s,1}), \ldots, (A_{s,t-1}, Y_{s,t-1}))$ denote the events in task $s$ upto and excluding round $t$ for all $t \geq 1$ ($H_{s,1} = \emptyset$). The events in task $s$ is denoted as $H_s = H_{s,n+1}$ and all the events upto and including stage $s$ is denoted as $H_{1:s} = \cup_{s'=1}^{s} H_{s'}$. Let us also define history upto and excluding round $t$ in task $s$ as $H_{1:s,t} = \{H_{1:s-1} \cup H_{s,t}\}$, with $H_{1:s} = H_{1:s,n+1}$. Given the history upto and excluding round $t$ in task $s$, the conditional probability is given as $\mathbb{P}_{s,t}(\cdot) = \mathbb{P}[\cdot \mid H_{1:s,t}]$, and the conditional expectation is given as $\mathbb{E}_{s,t}(\cdot) = \mathbb{E}[\cdot \mid H_{1:s,t}]$. Note $\mathbb{P}[\cdot]$ and $\mathbb{E}[\cdot]$ denote the unconditional probability and expectation, respectively.

**History dependent Entropy and Mutual Information:** We now define the entropy and mutual information terms as a function of history.

The mutual information between the parameter $\theta_{s,*}$, and the action $(A_{s,t})$ and reward $(Y_{s,t})$ at the beginning of round $t$ in task $s$, for any $s \leq m$ and $t \leq n$, as a function of history is defined as

$$I_{s,t}(\theta_{s,*}; A_{s,t}, Y_{s,t}) = \mathbb{E}_{s,t}\left[\log\left(\frac{\mathbb{P}_{s,t}(\theta_{s,*}, Y_{s,t}, A_{s,t})}{\mathbb{P}_{s,t}(\theta_{s,*})\mathbb{P}_{s,t}(Y_{s,t}, A_{s,t})}\right)\right]$$

We also define the mutual information between the parameter $\mu_*$, and the action $(A_{s,t})$ and reward $(Y_{s,t})$ at the beginning of round $t$ in task $s$, for any $s \leq m$ and $t \leq n$ as

$$I_{s,t}(\mu_*; A_{s,t}, Y_{s,t}) = \mathbb{E}_{s,t}\left[\log\left(\frac{\mathbb{P}_{s,t}(\mu_*, Y_{s,t}, A_{s,t})}{\mathbb{P}_{s,t}(\mu_*)\mathbb{P}_{s,t}(Y_{s,t}, A_{s,t})}\right)\right]$$

Further, the history dependent conditional mutual information between $(\mu_*, \theta_{s,*})$, and $A_{s,t}$ and $Y_{s,t}$, namely $I_{s,t}(\theta_{s,*}, \mu_*; A_{s,t}, Y_{s,t})$, is defined below.

$$I_{s,t}(\theta_{s,*}, \mu_*; A_{s,t}, Y_{s,t}) = \mathbb{E}_{s,t}\left[\log\left(\frac{\mathbb{P}_{s,t}(\theta_{s,*}, \mu_*, Y_{s,t}, A_{s,t})}{\mathbb{P}_{s,t}(\theta_{s,*}, \mu_*)\mathbb{P}_{s,t}(Y_{s,t}, A_{s,t})}\right)\right]$$

Finally, we define the history dependent conditional mutual information between $\theta_{s,*}$, and $A_{s,t}$ and $Y_{s,t}$ given $\mu_*$ as $I_{s,t}(\theta_{s,*}; A_{s,t}, Y_{s,t} \mid \mu_*)$.

$$I_{s,t}(\theta_{s,*}; A_{s,t}, Y_{s,t} \mid \mu_*) = \mathbb{E}_{s,t}\left[\log\left(\frac{\mathbb{P}_{s,t}(\theta_{s,*}, Y_{s,t}, A_{s,t} \mid \mu_*)}{\mathbb{P}_{s,t}(\theta_{s,*} \mid \mu_*)\mathbb{P}_{s,t}(Y_{s,t}, A_{s,t} \mid \mu_*)}\right)\right]$$

The conditional entropy terms are defined as follows:

$$h_{s,t}(\theta_{s,*}) = \mathbb{E}_{s,t}\left[-\log\left(\mathbb{P}_{s,t}(\theta_{s,*})\right)\right],$$
$$h_{s,t}(\mu_*) = \mathbb{E}_{s,t}\left[-\log\left(\mathbb{P}_{s,t}(\mu_*)\right)\right],$$
$$h_{s,t}(\theta_{s,*} \mid \mu_*) = \mathbb{E}_{s,t}\left[-\log\left(\mathbb{P}_{s,t}(\theta_{s,*} \mid \mu_*)\right)\right].$$

Therefore, all the different mutual information terms $I_{s,t}(\cdot; A_{s,t}, Y_{s,t})$, and the entropy terms $h_{s,t}(\cdot)$ are random variables that depends on the history $H_{1:s,t}$.

We next state some entropy and mutual information relationships which we will use later.

**Proposition 8.** *For all $s$, $t$, and any history $H_{1:s,t}$, the following hold*

$$I_{s,t}(\theta_{s,*}, \mu_*; A_{s,t}, Y_{s,t}) = I_{s,t}(\mu_*; A_{s,t}, Y_{s,t}) + I_{s,t}(\theta_{s,*}; A_{s,t}, Y_{s,t} \mid \mu_*),$$
$$I_{s,t}(\theta_{s,*}; A_{s,t}, Y_{s,t}) = h_{s,t}(\theta_{s,*}) - h_{s,t+1}(\theta_{s,*}).$$

**History Independent Entropy and Mutual Information:** The history independent conditional mutual information and entropy terms are then given by taking expectation over the possible histories

$$I(\cdot; A_{s,t}, Y_{s,t} \mid H_{1:s,t}) = \mathbb{E}[I_{s,t}(\cdot; A_{s,t}, Y_{s,t})], \quad h(\cdot \mid H_{1:s,t}) = \mathbb{E}[h_{s,t}(\cdot)]$$
$$I(\cdot; A_{s,t}, Y_{s,t} \mid \mu_*, H_{1:s,t}) = \mathbb{E}[I_{s,t}(\cdot; A_{s,t}, Y_{s,t} \mid \mu_*)], \quad h(\cdot \mid \mu_*, H_{1:s,t}) = \mathbb{E}[h_{s,t}(\cdot \mid \mu_*)]$$

An important quantity that will play a pivotal role in our regret decomposition is the conditional mutual information of the meta-parameter given the entire history, which is expressed as

$$I(\mu_*; H_{1:m}) = \sum_{s=1}^{m} \sum_{t=1}^{n} I(\mu_*; A_{s,t}, Y_{s,t} \mid H_{1:s,t}) = \mathbb{E} \sum_{s=1}^{m} \sum_{t=1}^{n} I_{s,t}(\mu_*; A_{s,t}, Y_{s,t}).$$

The first equality is due to chain rule of mutual information, where at each round the new history $H_{1:s,t+1} = H_{1:s,t} \cup (A_{s,t}, Y_{s,t})$.

Similarly, in each stage $s$, the mutual information between parameter $\theta_{s,*}$ and the events in stage $s$, i.e. $H_s$, conditioned on $\mu_*$ and history up to task $(s-1)$ is key in quantifying the local regret of task $s$. Which is again expressed as

$$I(\theta_{s,*}; H_s \mid \mu_*, H_{1:s-1}) = \sum_{t=1}^{n} I(\theta_{s,*}; A_{s,t}, Y_{s,t} \mid \mu_*, H_{1:s,t}) = \mathbb{E} \sum_{t=1}^{n} I_{s,t}(\theta_{s,*}; A_{s,t}, Y_{s,t} \mid \mu_*).$$

The first inequality again follows chain rule of mutual information with new history being the combination of old history, and the action and the observed reward in the current round.

We further have the relation of mutual information and conditional entropy as

$$I(\theta_{s,*}; H_s \mid \mu_*, H_{1:s-1}) = h(\theta_{s,*} \mid \mu_*, H_{1:s-1}) - h(\theta_{s,*} \mid \mu_*, H_{1:s}),$$
$$I(\mu_*; H_{1:m}) = h(\mu_*) - h(\mu_* \mid H_{1:m}).$$

**Weyl's Inequalities:** In this paper, the matrices under consideration are all Positive Semi-definite (PSD) and symmetric. Thus, the eignevalues are non-negative and admits a total order. We denote the eigenvalues of a PSD matrix $A \in \mathbb{R}^d$, for any integer $d \geq 1$, as $\lambda_d(A) \leq \cdots \leq \lambda_1(A)$; where $\lambda_1(A)$ is the maximum eigenvalue, and $\lambda_d(A)$ is the minimum eigenvalue of the PSD matrix $A$.

Weyl's inequality states for two Hermitian matrices (PSD and Symmetric in reals) $A$ and $B$,

$$\lambda_j(A) + \lambda_k(B) \leq \lambda_i(A+B) \leq \lambda_r(A) + \lambda_s(B), \quad \forall \, j + k - d \geq i \geq r + s - 1.$$

The two important relations, derived from Weyl's inequality, that we frequently use in the proofs are given next. For PSD and symmetric matrices $\{A_i\}$ we have

$$\lambda_1(\sum_i A_i) \leq \sum_i \lambda_1(A_i), \quad \text{and } \lambda_d(\sum_i A_i) \geq \sum_i \lambda_d(A_i).$$

**Lemma 2.** *Suppose that* (3) *holds for all* $s \in [m]$ *and* $t \in [n]$, *for some* $\Gamma_{s,t}, \epsilon_{s,t} \geq 0$. *In addition, let* $(\Gamma_s)_{s \in [m]}$ *and* $\Gamma$ *be non-negative constants such that* $\Gamma_{s,t} \leq \Gamma_s \leq \Gamma$ *holds for all* $s \in [m]$ *and* $t \in [n]$ *almost surely. Then*

$$R(m,n) \leq \Gamma \sqrt{mnI(\mu_*; H_{1:m})} + \sum_{s=1}^{m} \Gamma_s \sqrt{nI(\theta_{s,*}; H_s \mid \mu_*, H_{1:s-1})} + \sum_{s=1}^{m}\sum_{t=1}^{n} \mathbb{E}\left[\epsilon_{s,t}\right].$$

*Proof.* The proof follows through the series of inequalities below (explanation added).

$$R(m,n) = \mathbb{E}\sum_{s,t}[\Delta_{s,t}]$$

$$[\text{Eq. (3)}] \leq \mathbb{E}\sum_{s,t}\Gamma_{s,t}\sqrt{I_{s,t}(\theta_{s,*}; A_{s,t}, Y_{s,t})} + \mathbb{E}\sum_{s,t}\epsilon_{s,t}$$

$$[I(X;Z) \leq I(X,Y;Z)] \leq \mathbb{E}\sum_{s,t}\Gamma_{s,t}\sqrt{I_{s,t}(\theta_{s,*}, \mu_*; A_{s,t}, Y_{s,t})} + \mathbb{E}\sum_{s,t}\epsilon_{s,t}$$

$$[\text{Chain Rule}] = \mathbb{E}\sum_{s,t}\Gamma_{s,t}\sqrt{I_{s,t}(\mu_*; A_{s,t}, Y_{s,t}) + I_{s,t}(\theta_{s,*}; A_{s,t}, Y_{s,t} \mid \mu_*)} + \mathbb{E}\sum_{s,t}\epsilon_{s,t}$$

$$[\sqrt{a+b} \leq \sqrt{a} + \sqrt{b}] \leq \mathbb{E}\sum_{s,t}\Gamma_{s,t}\sqrt{I_{s,t}(\mu_*; A_{s,t}, Y_{s,t})} + \mathbb{E}\sum_{s,t}\Gamma_{s,t}\sqrt{I_{s,t}(\theta_{s,*}; A_{s,t}, Y_{s,t} \mid \mu_*)}$$

$$+ \mathbb{E}\sum_{s,t}\epsilon_{s,t}$$

$$[\Gamma_{s,t} \leq \Gamma_s \leq \Gamma, \forall s,t, \text{ w.p. } 1] \leq \Gamma\,\mathbb{E}\sum_{s,t}\sqrt{I_{s,t}(\mu_*; A_{s,t}, Y_{s,t})} + \sum_s \Gamma_s\left[\mathbb{E}\sum_t \sqrt{I_{s,t}(\theta_{s,*}; A_{s,t}, Y_{s,t} \mid \mu_*)}\right]$$

$$+ \mathbb{E}\sum_{s,t}\epsilon_{s,t}$$

$$[\text{Jensen's Inequality}] \leq \Gamma\sum_{s,t}\sqrt{\mathbb{E}I_{s,t}(\mu_*; A_{s,t}, Y_{s,t})} + \sum_s \Gamma_s \sum_t \sqrt{\mathbb{E}I_{s,t}(\theta_{s,*}; A_{s,t}, Y_{s,t} \mid \mu_*)}$$

$$+ \mathbb{E}\sum_{s,t}\epsilon_{s,t}$$

$$[\text{Cauchy-Schwarz}] \leq \Gamma\sqrt{mn\sum_{s,t}\mathbb{E}I_{s,t}(\mu_*; A_{s,t}, Y_{s,t})} + \sum_s \Gamma_s \sqrt{n\sum_t \mathbb{E}I_{s,t}(\theta_{s,*}; A_{s,t}, Y_{s,t} \mid \mu_*)}$$

$$+ \mathbb{E}\sum_{s,t}\epsilon_{s,t}$$

$$[\text{Chain Rule}] = \Gamma\sqrt{mnI(\mu_*; H_{1:m})} + \sum_s \Gamma_s \sqrt{nI(\theta_{s,*}; H_s \mid \mu_*, H_{1:s-1})} + \mathbb{E}\sum_{s,t}\epsilon_{s,t}$$

- The first inequality follows due to Eq. (3).

- The second inequality uses the fact that $I(X;Z) \leq I(X,Y;Z)$ for any random variables $X, Y$, and $Z$. Here $X = \theta_{s,*}$, $Y = \mu_*$, and $Z = (A_{s,t}, Y_{s,t})$.

- The second equality uses the chain rule $I(X,Y;Z) = I(X;Z) + I(X;Z \mid Y)$, as stated in Proposition 8, with the same random variables $X, Y$, and $Z$.

- The Jensen's inequality uses concavity of $\sqrt{\cdot}$.

$\square$

# C Proofs for Section 4.2: Linear Bandit

## C.1 Marginalization of the Variables

**Notation in Marginalization:** Let $\mathcal{N}(x; \mu, \Sigma)$ denote a (possibly multivariate) Gaussian p.d.f. with mean $\mu$ and covariance matrix $\Sigma$ for variable $x$. We now recall the notations of posterior distributions at different time of our algorithm

$$P(\theta; \mu) = \mathbb{P}\left(\theta_{s,*} = \theta \mid \mu_* = \mu\right) = \mathcal{N}(\theta; \mu, \Sigma_0), \quad Q(\mu) = \mathbb{P}\left(\mu_* = \mu\right) = \mathcal{N}(\mu; \mu_0, \Sigma_q)$$

$$P_s(\theta) = \mathbb{P}\left(\theta_{s,*} = \theta \mid H_{1:s-1}\right) = \int_\mu P(\theta; \mu) Q_s(\mu)\, \mathrm{d}\mu,$$

$$P_{s,t}(\theta) = \mathbb{P}\left(\theta_{s,*} = \theta \mid H_{1:s,t}\right) \propto \mathbb{P}\left(H_{s,t} \mid \theta_{s,*} = \theta\right) P_s(\theta),$$

$$Q_s(\mu) = \mathbb{P}\left(\mu_* = \mu \mid H_{1:s-1}\right) = \int_\theta \mathbb{P}\left(H_{s-1} \mid \theta_{s-1,*} = \theta\right) P(\theta; \mu)\, \mathrm{d}\theta Q_{s-1}(\mu)$$

The marginalization is proved in an inductive manner due to the dependence of the action matrix $A$ on the history. We recall the expression of the rewards,

$$Y_{s,t} = A_{s,t}^T \theta_{s,*} + w_{s,t}$$

In each round $t$ and task $s$, given the parameter $\theta_{s,*}$ and the action $A_{s,t}$, the reward $Y_{s,t}$ has the p.d.f. $\mathbb{P}(Y_{s,t} \mid \theta_{s,*}, A_{s,t}) = \mathcal{N}(Y_{s,t}; A_{s,t}^T \theta_{s,*}, \sigma^2)$. Let $\propto_X$ denote that the proportionality constant is independent of $X$ (possibly a set).

We obtain the posterior probability of the true parameter in task $s$ in round $t$, given the true parameter $\mu_*$. Let us define for all $s \le m$, and $t \le n$.

$$P_{s,t,\mu_*}(\theta) = \mathbb{P}(\theta_{s,*} = \theta \mid \mu_*, H_{1:s,t}) \propto \prod_{t'=1}^{t-1} \mathbb{P}(Y_{s,t'} \mid \theta_{s,*} = \theta, A_{s,t'}) P(\theta, \mu_*)$$

$$\propto_\theta \prod_{t'=1}^{t-1} \exp\left(-\frac{(Y_{s,t'} - A_{s,t'}^T \theta)^2}{2\sigma^2}\right) \mathcal{N}(\theta; \mu_*, \Sigma_0)$$

$$\propto_\theta \exp\left(-\sum_{t'=1}^{t-1} (\theta - A_{s,t'} Y_{s,t'})^T \frac{A_{s,t'} A_{s,t'}^T}{2\sigma^2} (\theta - A_{s,t'} Y_{s,t'})\right) \mathcal{N}(\theta; \mu_*, \Sigma_0)$$

$$\propto_\theta \exp\left(-\left(\theta - \bar{\theta}\right)^T \sum_{t'=1}^{t-1} \frac{A_{s,t'} A_{s,t'}^T}{2\sigma^2} \left(\theta - \bar{\theta}\right)\right) \mathcal{N}(\theta; \mu_*, \Sigma_0) \quad \left[\bar{\theta} = \left(\sum_{t'=1}^{t-1} \frac{A_{s,t'} A_{s,t'}^T}{\sigma^2}\right)^{-1} \sum_{t'=1}^{t-1} A_{s,t'} Y_{s,t'}\right]$$

$$\propto_\theta \mathcal{N}\left(\theta; \hat{\Sigma}_{s,t,\mu_*}\left(\Sigma_0^{-1} \mu_* + \sum_{t'=1}^{t-1} A_{s,t'} Y_{s,t'}\right), \hat{\Sigma}_{s,t,\mu_*}\right) \quad \left[\hat{\Sigma}_{s,t,\mu_*}^{-1} = \Sigma_0^{-1} + \sum_{t'=1}^{t-1} \frac{A_{s,t'} A_{s,t'}^T}{\sigma^2}\right]$$

We now obtain the posterior probability of the true parameter in task $s$ in round $t$ as by taking integral over the prior of the parameter $\mu_*$.

$$P_{s,t}(\theta) = \mathbb{P}(\theta_{s,*} = \theta \mid H_{1:s,t}) \propto \prod_{t'=1}^{t-1} \mathbb{P}(Y_{s,t'} \mid \theta_{s,*} = \theta, A_{s,t'}) \int_\mu P(\theta, \mu) Q_s(\mu) d\mu$$

$$\propto_\theta \prod_{t'=1}^{t-1} \exp\left(-\frac{(Y_{s,t'} - A_{s,t'}^T \theta)^2}{2\sigma^2}\right) \int_\mu \mathcal{N}(\theta; \mu, \Sigma_0) \mathcal{N}(\mu; \hat{\mu}_s, \hat{\Sigma}_s) d\mu$$

$$\propto_\theta \exp\left(-\sum_{t'=1}^{t-1} (\theta - A_{s,t'} Y_{s,t'})^T \frac{A_{s,t'} A_{s,t'}^T}{2\sigma^2} (\theta - A_{s,t'} Y_{s,t'})\right) \mathcal{N}(\theta; \hat{\mu}_s, \Sigma_0 + \hat{\Sigma}_s)$$

$$\propto_\theta \exp\left(-\sum_{t'=1}^{t-1} (\theta - A_{s,t'} Y_{s,t'})^T \frac{A_{s,t'} A_{s,t'}^T}{2\sigma^2} (\theta - A_{s,t'} Y_{s,t'})\right) \mathcal{N}(\theta; \hat{\mu}_s, \Sigma_0 + \hat{\Sigma}_s)$$

$$\propto_\theta \exp\left(-(\theta - \bar{\theta})^T \sum_{t'=1}^{t-1} \frac{A_{s,t'} A_{s,t'}^T}{2\sigma^2} (\theta - \bar{\theta})\right) \mathcal{N}(\theta; \hat{\mu}_s, \Sigma_0 + \hat{\Sigma}_s) \quad \left[\bar{\theta} = (\sum_{t'=1}^{t-1} \frac{A_{s,t'} A_{s,t'}^T}{\sigma^2})^{-1} \sum_{t'=1}^{t-1} A_{s,t'} Y_{s,t'}\right]$$

$$\propto_\theta \mathcal{N}\left(\theta; (\sum_{t'=1}^{t-1} \frac{A_{s,t'} A_{s,t'}^T}{\sigma^2})^{-1} \sum_{t'=1}^{t-1} A_{s,t'} Y_{s,t'}, (\sum_{t'=1}^{t-1} \frac{A_{s,t'} A_{s,t'}^T}{\sigma^2})^{-1}\right) \mathcal{N}(\theta; \hat{\mu}_s, \Sigma_0 + \hat{\Sigma}_s)$$

$$\propto_\theta \mathcal{N}\left(\theta; \hat{\Sigma}_{s,t}\left((\Sigma_0 + \hat{\Sigma}_s)^{-1} \hat{\mu}_s + \sum_{t'=1}^{t-1} A_{s,t'} Y_{s,t'}\right), \hat{\Sigma}_{s,t}\right) \quad \left[\hat{\Sigma}_{s,t}^{-1} = (\Sigma_0 + \hat{\Sigma}_s)^{-1} + \sum_{t'=1}^{t-1} \frac{A_{s,t'} A_{s,t'}^T}{\sigma^2}\right]$$

Thus, for $\hat{\mu}_{s,t} = \hat{\Sigma}_{s,t}\left((\Sigma_0 + \hat{\Sigma}_s)^{-1}\hat{\mu}_s + \sum_{t'=1}^{t-1} A_{s,t'} Y_{s,t'}\right)$, the parameter conditioned on the history is distributed as $\theta_{s,*} \mid H_{1:s,t} \sim \mathcal{N}(\hat{\mu}_{s,t}, \hat{\Sigma}_{s,t})$.

We now compute the posterior of the meta-parameter $\mu_*$ in a similar way, but some of the computation can be avoided by using Lemma 7.

$$Q_{s+1}(\mu) = \int_\theta \mathbb{P}(H_s \mid \theta_{s,*} = \theta) P(\theta; \mu) d\theta Q_s(\mu)$$

$$\propto_{\theta,\mu} \int_\theta \prod_{t=1}^n \mathbb{P}(Y_{s,t} \mid \theta_{s,*} = \theta, A_{s,t}) P(\theta, \mu) d\theta Q_s(\mu)$$

$$\propto_{\theta,\mu} \prod_{\ell=1}^s \int_{\theta_\ell} \prod_{t=1}^n \mathbb{P}(Y_{\ell,t} \mid \theta_{\ell,*} = \theta_\ell, A_{\ell,t}) P(\theta_\ell, \mu) d\theta_s Q_0(\mu)$$

$$= \mathcal{N}(\hat{\mu}_{s+1}, \hat{\Sigma}_{s+1})$$

The second equality is obtained by expanding out the $Q_s(\mu)$ expressions iteratively, and using the fact that $Q_0(\mu)$ is the prior distribution of $\mu$ at the beginning. The final equality follows from the application of Lemma 7, by observing that the expression describes a setting identical to the setting therein, with actions $x_{\ell,t} = A_{\ell,t}$ for all $\ell \in [s]$ and $t \in [n]$. The probability of playing the actions $A_{\ell,t}$ (as opposed to fixed $x_{\ell,t}$ in Lemma 7) are absorbed by the proportionality constant.

Recall that we have due to Lemma 7, for $G_\ell = \sum_{t=1}^n A_{\ell,t} A_{\ell,t}^T$, $\forall \ell \in [m]$ and for any $s \in [m]$,

$$\hat{\Sigma}_s^{-1} = \Sigma_q^{-1} + \sum_{\ell=1}^{s-1} \frac{G_\ell}{\sigma^2} - \frac{G_\ell}{\sigma^2}\left(\Sigma_0^{-1} + \frac{G_\ell}{\sigma^2}\right)^{-1} \frac{G_\ell}{\sigma^2} = \Sigma_q^{-1} + \sum_{\ell=1}^{s-1} \frac{G_\ell}{\sigma^2}\left(\Sigma_0^{-1} + \frac{G_\ell}{\sigma^2}\right)^{-1} \Sigma_0^{-1}.$$

Further, if in task $\ell$ if forced exploration is used, then $G_\ell$ is invertible, and using Woodbury matrix identity we have

$$\frac{G_\ell}{\sigma^2}\left(\Sigma_0^{-1} + \frac{G_\ell}{\sigma^2}\right)^{-1} \Sigma_0^{-1} = \left(\Sigma_0 + (\frac{G_\ell}{\sigma^2})^{-1}\right)^{-1}.$$

## C.2 Proof of Lemma 4

**Lemma 4.** *For any $H_{1:s,t}$-adapted action sequence and any $s \in [m]$, we have*

$$I(\theta_{s,*}; H_s \mid \mu_*, H_{1:s-1}) \leq \frac{d}{2} \log\left(1 + \frac{\lambda_1(\Sigma_0)n}{\sigma^2}\right), \quad I(\mu_*; H_{1:m}) \leq \frac{d}{2} \log\left(1 + \frac{\lambda_1(\Sigma_q)m}{\lambda_d(\Sigma_0) + \sigma^2/n}\right).$$

*Proof.* We obtain the conditional mutual entropy of $\theta_{s,*}$ given the history upto $(s-1)$-th task and $\theta$ (similar to Lu et al.[33])

$$
\begin{aligned}
I(\theta_{s,*}; H_s \mid \mu_*, H_{1:s-1}) &= h(\theta_{s,*} \mid \mu_*, H_{1:s-1}) - h(\theta_{s,*} \mid \mu_*, H_{1:s}) \\
&= \mathbb{E}[h_{s-1,n+1}(\theta_{s,*} \mid \mu_*)] - \mathbb{E}[h_{s,n+1}(\theta_{s,*} \mid \mu_*)] \\
&= \frac{1}{2} \log(\det(2\pi e \Sigma_0)) - \mathbb{E}[\frac{1}{2} \log(\det(2\pi e \hat{\Sigma}_{s,n,\mu_*}))] \\
&= \frac{1}{2} \mathbb{E}[\log(\det(\Sigma_0) \det(\hat{\Sigma}_{s,n,\mu_*}^{-1}))] \\
&= \frac{1}{2} \mathbb{E}\left[\prod_{i=1}^{d} \lambda_i(\Sigma_0)\lambda_i(\hat{\Sigma}_{s,n,\mu_*}^{-1}))\right] \\
&\leq \frac{1}{2} \log\left(\prod_{i=1}^{d} \lambda_i(\Sigma_0)\left(\frac{1}{\lambda_i(\Sigma_0)} + \frac{n}{\sigma^2}\right)\right) \\
&\leq \frac{d}{2} \log\left(1 + n\frac{\lambda_1(\Sigma_0)}{\sigma^2}\right)
\end{aligned}
$$

The first inequality follows from the definition of conditional mutual information (here we have outer expectation). Using the relation between the history-independent and history-dependent entropy terms we obtain the second inequality. Note that $h_{s-1,n+1}(\theta_{s,*} \mid \mu_*)$ is independent of history, as the $\theta_{s,*}$ given $\mu_*$ does not depend on old tasks.

For the first inequality, we derive the following history independent bound.

$$
\begin{aligned}
\lambda_i(\hat{\Sigma}_{s,n,\mu_*}^{-1}) &= \lambda_i\left(\Sigma_0^{-1} + \frac{1}{\sigma^2} \sum_{t'=1}^{n} A_{s,t'} A_{s,t'}^T\right) \\
&\leq \lambda_i(\Sigma_0^{-1}) + \lambda_1\left(\sum_{t'=1}^{n} \frac{A_{s,t'} A_{s,t'}^T}{\sigma^2}\right) \\
&\leq \frac{1}{\lambda_i(\Sigma_0)} + tr\left(\sum_{t'=1}^{n} \frac{A_{s,t'} A_{s,t'}^T}{\sigma^2}\right) \\
&\leq \frac{1}{\lambda_i(\Sigma_0)} + \frac{n}{\sigma^2}
\end{aligned}
$$

The matrices $\frac{1}{\sigma^2} \sum_{t'=1}^{n} A_{s,t'} A_{s,t'}^T$, and $\Sigma_0^{-1}$ are Hermitian matrices, giving us the first inequality by applicaiton of Weyl's inequality. The last inequality first uses linearity of trace, and $tr(A_{s,t'} A_{s,t'}^T) = tr(A_{s,t'}^T A_{s,t'}) \leq 1$, by Assumption 1.

Similarly, we derive the mutual information of the meta-parameter of $\theta$ given the history as follows

$$
\begin{aligned}
I(\mu_*; H_{1:m}) &= h(\mu_*) - h(\mu_* \mid H_{1:m}) \\
&= h(\mu_*) - \mathbb{E}[h_{m,n+1}(\mu_*)] \\
&= \frac{1}{2} \log(\det(2\pi e \Sigma_q)) - \mathbb{E}[\frac{1}{2} \log(\det(2\pi e \hat{\Sigma}_{m+1}))] \\
&= \frac{1}{2} \mathbb{E}[\log(\det(\Sigma_q) \det(\hat{\Sigma}_{m+1}^{-1}))] \\
&\leq \frac{d}{2} \log\left(1 + \frac{mn\lambda_1(\Sigma_q)}{n\lambda_d(\Sigma_0) + \sigma^2}\right)
\end{aligned}
$$

For the final inequality above, we derive a history independent bounds in a similar manner.

$$\lambda_i(\hat{\Sigma}_{m+1}^{-1}) \leq \lambda_i(\Sigma_q^{-1}) + \lambda_1\left(\sum_{s'=1}^{m}\left(\Sigma_0 + (\sum_{t'=1}^{n} \frac{A_{s',t'} A_{s',t'}^T}{\sigma^2})^{-1}\right)^{-1}\right)$$

$$\leq \lambda_i(\Sigma_q^{-1}) + \sum_{s'=1}^{m} \lambda_1 \left( \left( \Sigma_0 + (\sum_{t'=1}^{n} \frac{A_{s',t'} A_{s',t'}^T}{\sigma^2})^{-1} \right)^{-1} \right)$$

$$\leq \frac{1}{\lambda_i(\Sigma_q)} + \sum_{s'=1}^{m} \lambda_d^{-1} \left( \Sigma_0 + (\sum_{t'=1}^{n} \frac{A_{s',t'} A_{s',t'}^T}{\sigma^2})^{-1} \right)$$

$$\leq \frac{1}{\lambda_i(\Sigma_q)} + \sum_{s'=1}^{m} \left( \lambda_d(\Sigma_0) + \lambda_d \left( (\sum_{t'=1}^{n} \frac{A_{s',t'} A_{s',t'}^T}{\sigma^2})^{-1} \right) \right)^{-1}$$

$$\leq \frac{1}{\lambda_i(\Sigma_q)} + \sum_{s'=1}^{m} \left( \lambda_d(\Sigma_0) + \lambda_1^{-1}(\sum_{t'=1}^{n} \frac{A_{s',t'} A_{s',t'}^T}{\sigma^2}) \right)^{-1}$$

$$\leq \frac{1}{\lambda_i(\Sigma_q)} + \sum_{s'=1}^{m} \left( \lambda_d(\Sigma_0) + \frac{\sigma^2}{n} \right)^{-1} = \frac{1}{\lambda_i(\Sigma_q)} + \frac{mn}{n\lambda_d(\Sigma_0) + \sigma^2}$$

$\square$

### C.3 Proof of Lemma 3

**Lemma 3.** *For all tasks $s \in [m]$, rounds $t \in [n]$, and any $\delta \in (0,1]$, (3) holds almost surely for*

$$\Gamma_{s,t} = 4\sqrt{\frac{\sigma_{\max}^2(\hat{\Sigma}_{s,t})}{\log(1 + \sigma_{\max}^2(\hat{\Sigma}_{s,t})/\sigma^2)} \log(4|\mathcal{A}|/\delta)}, \quad \epsilon_{s,t} = \sqrt{2\delta\sigma_{\max}^2(\hat{\Sigma}_{s,t})} + 2\mathcal{E}_{s,t}\mathbb{E}_{s,t}[\|\theta_{s,*}\|_2],$$

*where $\mathcal{E}_{s,t}$ is the indicator of forced exploration in round $t$ of task $s$. Moreover, for each task $s$, the following history-independent bound holds almost surely,*

$$\sigma_{\max}^2(\hat{\Sigma}_{s,t}) \leq \lambda_1(\Sigma_0) \left( 1 + \frac{\lambda_1(\Sigma_q)\left(1 + \frac{\sigma^2}{\eta\lambda_1(\Sigma_0)}\right)}{\lambda_1(\Sigma_0) + \sigma^2/\eta + s\lambda_1(\Sigma_q)} \right). \tag{5}$$

*Proof.* We next derive the confidence interval bounds, similar to Lu et al. [33], for the reward $Y_{s,t}$ around it's mean conditioned on the history $H_{s-1} \cup H_{s,t-1}$. Let $\hat{\theta}_{s,t}$ be the parameter sampled by TS in task $s$ and round $t$, when we do not have forced exploration.

$$\mathbb{E}_{s,t}[\Delta_{s,t}] = \mathbb{E}_{s,t}[A_{s,*}^T \theta_{s,*} - A_{s,t}^T \theta_{s,*}] = \mathbb{E}_{s,t}[A_{s,t}^T \hat{\theta}_{s,t} - A_{s,t}^T \theta_{s,*}]$$

The last equality holds as for Thompson sampling ($\stackrel{d}{=}$ denotes equal distribution)

$$A_{s,*}^T \theta_{s,*} \mid H_{1:s,t} \stackrel{d}{=} A_{s,t}^T \hat{\theta}_{s,t} \mid H_{1:s,t}.$$

When for task $s$ and round $t$ we have forced exploration the bound is given as

$$\mathbb{E}_{s,t}[\Delta_{s,t}] = \mathbb{E}_{s,t}[A_{s,t}^T \hat{\theta}_{s,t} - A_{s,t}^T \theta_{s,*}] + \mathbb{E}_{s,t}[A_{s,*}^T \theta_{s,*} - A_{s,t}^T \hat{\theta}_{s,t}]$$
$$\leq \mathbb{E}_{s,t}[A_{s,t}^T \hat{\theta}_{s,t} - A_{s,t}^T \theta_{s,*}] + 2\mathbb{E}_{s,t}[\max_{a \in \mathcal{A}} |a^T \theta_{s,*}|]$$
$$\leq \mathbb{E}_{s,t}[A_{s,t}^T \hat{\theta}_{s,t} - A_{s,t}^T \theta_{s,*}] + 2\mathbb{E}_{s,t}[\|\theta_{s,*}\|_2].$$

In the second last inequality we use the fact that $\theta_{s,*} \mid H_{1:s,t} \stackrel{d}{=} \hat{\theta}_{s,t} \mid H_{1:s,t}$.

Recall $Y_{s,t}(a)$ denote the reward obtained by taking action $a$ in task $s$ and round $t$. Also recall that $\hat{\theta}_{s,t} \mid H_{1:s,t} \sim \mathcal{N}(\hat{\mu}_{s,t}, \hat{\Sigma}_{s,t})$. Let us consider the set

$$\Theta_{s,t} = \{\theta :| a^T \theta - a^T \hat{\theta}_{s,t} |\leq \frac{\Gamma_{s,t}}{2} \sqrt{I_{s,t}(\theta_{s,*}; a, Y_{s,t}(a))}, \forall a \in \mathcal{A}\}.$$

The history dependent conditional mutual entropy of $\theta_{s,*}$ given the history $H_{s-1} \cup H_{s,t}$ (not $\mu_*$) (which will be useful in deriving concentration bounds) as

$$
\begin{aligned}
I_{s,t}(\theta_{s,*}; A_{s,t}, Y_{s,t}) &= h_{s,t}(\theta_{s,*}) - h_{s,t+1}(\theta_{s,*}) \\
&= \tfrac{1}{2} \log(\det(2\pi e(\hat{\Sigma}_{s,t-1}))) - \tfrac{1}{2} \log(\det(2\pi e\hat{\Sigma}_{s,t})) \\
&= \tfrac{1}{2} \log(\det(\hat{\Sigma}_{s,t-1}\hat{\Sigma}_{s,t}^{-1})) \\
&= \tfrac{1}{2} \log\left(\det\left(I + \hat{\Sigma}_{s,t-1}\frac{A_{s,t}A_{s,t}^T}{\sigma^2}\right)\right) \\
&= \tfrac{1}{2} \log\left(\det\left(1 + \frac{A_{s,t}^T\hat{\Sigma}_{s,t-1}A_{s,t}}{\sigma^2}\right)\right)
\end{aligned}
$$

The last step above uses Matrix determinant lemma.[1] Recall that $\sigma_{\max}^2(\hat{\Sigma}_{s,t}) = \max_{a \in \mathcal{A}} a^T \hat{\Sigma}_{s,t} a$ for all $s \leq m$ and $t \leq n$. For $\delta \in (0,1]$, let

$$
\Gamma_{s,t} = 4\sqrt{\frac{\sigma_{\max}^2(\hat{\Sigma}_{s,t-1})}{\log(1 + \sigma_{\max}^2(\hat{\Sigma}_{s,t-1})/\sigma^2)} \log(\tfrac{4|\mathcal{A}|}{\delta})}.
$$

Now it follows from Lu et al. [33] Lemma 5 that for the $\Gamma_{s,t}$ defined as above we have

$$
\mathbb{P}_{s,t}(\hat{\theta}_{s,t} \in \Theta_{s,t}) \geq 1 - \delta/2.
$$

We continue with the regret decomposition as

$$
\mathbb{E}_{s,t}[\Delta_{s,t}]
$$
$$
= \mathbb{E}_{s,t}\left[\mathbb{1}(\hat{\theta}_{s,t}, \theta_{s,*} \in \Theta_{s,t})\left(A_{s,t}^T\hat{\theta}_{s,t} - A_{s,t}^T\theta_{s,*}\right)\right] + \mathbb{E}_{s,t}\left[\mathbb{1}^c(\hat{\theta}_{s,t}, \theta_{s,*} \in \Theta_{s,t})\left(A_{s,t}^T\hat{\theta}_{s,t} - A_{s,t}^T\theta_{s,*}\right)\right]
$$
$$
\leq \mathbb{E}_{s,t}\left[\sum_{a \in \mathcal{A}} \mathbb{1}(A_{s,t} = a)\Gamma_{s,t}\sqrt{I_{s,t}(\theta_{s,*}; a, Y_{s,t}(a))}\right]
$$
$$
+ \sqrt{\mathbb{P}_{s,t}(\hat{\theta}_{s,t} \text{ or } \theta_{s,*} \notin \Theta_{s,t})\mathbb{E}_{s,t}\left[\left(A_{s,t}^T\hat{\theta}_{s,t} - A_{s,t}^T\theta_{s,*}\right)^2\right]}
$$
$$
\leq \Gamma_{s,t}\sqrt{I_{s,t}(\theta_{s,*}; A_{s,t}Y_{s,t})} + \sqrt{\mathbb{P}_{s,t}(\hat{\theta}_{s,t} \text{ or } \theta_{s,*} \notin \Theta_{s,t})}\max_{a \in \mathcal{A}}\sqrt{\mathbb{E}_{s,t}\left[\left(a^T\hat{\theta}_{s,t} - a^T\theta_{s,*}\right)^2\right]}
$$
$$
\leq \Gamma_{s,t}\sqrt{I_{s,t}(\theta_{s,*}; A_{s,t}Y_{s,t})} + \underbrace{\sqrt{2\delta\sigma_{\max}^2(\hat{\Sigma}_{s,t-1})}}_{\epsilon_{s,t}}
$$

- The left side term in the first inequality uses the definition of $\Theta_{s,t}$. The right side term in the first inequality holds due to Cauchy–Schwarz. In particular, we use $\mathbb{E}[XY] \leq \sqrt{\mathbb{E}[X^2]\mathbb{E}[Y^2]}$ with $X = \mathbb{1}^c(\hat{\theta}_{s,t}, \theta_{s,*} \in \Theta_{s,t})$ and $Y = \left(A_{s,t}^T\hat{\theta}_{s,t} - A_{s,t}^T\theta_{s,*}\right)$.

- The left side term in the second inequality follows steps similar to proof of Lemma 3 in Lu et al. [33]. The right side term in the second inequality maximizes over the possible actions (we can take the max out of the expectation as action $A_{s,t}$ is a function of history upto task $s$, and round $t-1$). The last inequality follows from the following derivation

$$
\mathbb{E}_{s,t}\left[\left(a^T\hat{\theta}_{s,t} - a^T\theta_{s,*}\right)^2\right]
$$
$$
\leq \mathbb{E}_{s,t}\left[a^T\left((\hat{\theta}_{s,t} - \mu_{s,t-1}) - (\theta_{s,*} - \mu_{s,t-1})\right)^2\right]
$$
$$
\leq a^T\left(\mathbb{E}_{s,t}\left[(\hat{\theta}_{s,t} - \mu_{s,t-1})(\hat{\theta}_{s,t} - \mu_{s,t-1})^T\right] + \mathbb{E}_{s,t}\left[(\theta_{s,*} - \mu_{s,t-1})(\theta_{s,*} - \mu_{s,t-1})^T\right]\right)a
$$
$$
\leq 2a^T\hat{\Sigma}_{s,t-1}a \leq 2\sigma_{\max}^2(\hat{\Sigma}_{s,t-1})
$$

---

[1]Matrix determinant lemma states that for an invertible square matrix $A$, and vectors $u$ and $v$ $\det\left(A + uv^T\right) = \left(1 + v^TA^{-1}u\right)\det\left(A\right)$. We use $A = I$, $u = \hat{\Sigma}_{s,t-1}A_{s,t}$, and $v = A_{s,t}/\sigma^2$.

This conclude the proof of the first part.

We first claim that $\sigma^2_{\max}(\hat{\Sigma}_{s,t}) \leq \lambda_1(\hat{\Sigma}_{s,t})$. Indeed, as $\|a\|_2 \leq 1$, we have

$$\sigma^2_{\max}(\hat{\Sigma}_{s,t}) = \max_{a\in\mathcal{A}} a^T \hat{\Sigma}_{s,t} a \leq \max_{a\in\mathcal{A}} a^T \lambda_1(\hat{\Sigma}_{s,t}) a \leq \lambda_1(\hat{\Sigma}_{s,t}).$$

Furthermore, $\lambda_1(\hat{\Sigma}_{s,t})$ decreases with $s$ and $t$ (precisely with $n(s-1)+t$). To show this we use

$$\lambda_1(\hat{\Sigma}_{s,t}) = \lambda_d^{-1}(\hat{\Sigma}_{s,t}^{-1})$$

$$= \lambda_d^{-1}\left((\Sigma_0+\hat{\Sigma}_s)^{-1} + \sum_{t'=1}^{t} \frac{A_{s,t'}A_{s,t'}^T}{\sigma^2}\right)$$

$$\leq \lambda_d^{-1}\left((\Sigma_0+\hat{\Sigma}_s)^{-1} + \sum_{t'=1}^{t-1} \frac{A_{s,t'}A_{s,t'}^T}{\sigma^2}\right)$$

$$= \lambda_d^{-1}(\hat{\Sigma}_{s,t-1}^{-1}) = \lambda_1(\hat{\Sigma}_{s,t-1})$$

The inequality holds due to Weyl's inequality and $\frac{A_{s,t}A_{s,t}^T}{\sigma^2}$ being a PSD matrix. In particular, we have $\lambda_d(A+B) \geq \lambda_d(A) + \lambda_d(B)$, given $A$ and $B$ are Hermitian. Thus

$$\lambda_d^{-1}(A+B) \leq (\lambda_d(A) + \lambda_d(B))^{-1} \leq \lambda_d^{-1}(A).$$

Recall in each task $s$, due to forced exploration, we have $\lambda_d(\sum_{t'=1}^{n} \frac{A_{s,t'}A_{s,t'}^T}{\sigma^2}) \geq \frac{\eta}{\sigma^2}$, where $\eta$ is the forced exploration constant. We now prove an upper bound for the term $\lambda_1(\hat{\Sigma}_s)$ independent of action sequences.

$$\lambda_1(\Sigma_0 + \hat{\Sigma}_s) - \lambda_1(\Sigma_0) \leq \lambda_1(\hat{\Sigma}_s) = \lambda_d^{-1}(\hat{\Sigma}_s^{-1})$$

$$= \lambda_d^{-1}\left(\Sigma_q^{-1} + \sum_{s'=1}^{s-1}\left(\sum_{t'=1}^{n} \frac{A_{s',t'}A_{s',t'}^T}{\sigma^2}\right)\left(\Sigma_0^{-1} + \sum_{t'=1}^{n} \frac{A_{s',t'}A_{s',t'}^T}{\sigma^2}\right)^{-1}\Sigma_0^{-1}\right)$$

$$\leq \left(\lambda_d(\Sigma_q^{-1}) + \sum_{s'=1}^{s-1}\lambda_d\left(\left(\Sigma_0 + (\sum_{t'=1}^{n} \frac{A_{s',t'}A_{s',t'}^T}{\sigma^2})^{-1}\right)^{-1}\right)\right)^{-1}$$

$$\leq \left(\lambda_d(\Sigma_q^{-1}) + \sum_{s'=1}^{s-1}\left(\lambda_1(\Sigma_0) + \lambda_1\left((\sum_{t'=1}^{n} \frac{A_{s',t'}A_{s',t'}^T}{\sigma^2})^{-1}\right)\right)^{-1}\right)^{-1}$$

$$\leq \left(\lambda_1^{-1}(\Sigma_q) + s(\lambda_1(\Sigma_0) + \sigma^2/\eta)^{-1}\right)^{-1}$$

In the above derivation, we use the Weyl's inequalities multiple times. Note the direction of inequality should be $\leq$ if there are even number of inverses, whereas it should be $\geq$ if there are an odd number of inverses associated. The first inequality uses the inequality $\lambda_d(\sum_i A_i) \geq \sum_i \lambda_d(A_i)$ given all the matrices $A_i$-s are Hermitian. The second inequality similarly uses $\lambda_1(\sum_i A_i) \leq \sum_i \lambda_1(A_i)$ given all the matrices $A_i$-s are Hermitian. The final inequality uses the minimum eigenvalue bound when forced exploration is used.

This concludes the second part of the proof, in particular

$$\sigma^2_{\max}(\hat{\Sigma}_{s,t}) \leq \lambda_1(\Sigma_0 + \hat{\Sigma}_s) \leq \lambda_1(\Sigma_0)\left(1 + \frac{\lambda_1(\Sigma_q)(1+\frac{\sigma^2/\eta}{\lambda_1(\Sigma_0)})}{\lambda_1(\Sigma_0)+\sigma^2/\eta+s\lambda_1(\Sigma_q)}\right).$$

$\square$

### C.4 Proof of Theorem 5

**Theorem 5 (Linear bandit).** *The regret of* `AdaTS` *is bounded for any $\delta \in (0,1]$ as*

$$R(m,n) \leq \underbrace{c_1\sqrt{dmn}}_{\text{Learning of } \mu_*} + (m+c_2)\underbrace{R_\delta(n;\mu_*)}_{\text{Per-task regret}} + \underbrace{c_3 dm}_{\text{Forced exploration}},$$

*where*

$$c_1 = \sqrt{8\frac{\lambda_1(\Sigma_q)+\lambda_1(\Sigma_0)}{\log\left(1+\frac{\lambda_1(\Sigma_q)+\lambda_1(\Sigma_0)}{\sigma^2}\right)}\log(4|\mathcal{A}|/\delta)\log\left(1+\frac{\lambda_1(\Sigma_q)m}{\lambda_d(\Sigma_0)+\sigma^2/n}\right)},$$

$c_2 = \left(1+\frac{\sigma^2}{\eta\lambda_1(\Sigma_0)}\right)\log m$, *and* $c_3 = 2\sqrt{\|\mu_q\|_2^2+\mathrm{tr}(\Sigma_q+\Sigma_0)}$. *The* per-task regret *is bounded as* $R_\delta(n;\mu_*) \le c_4\sqrt{dn}+\sqrt{2\delta\lambda_1(\Sigma_0)}n$, *where*

$$c_4 = \sqrt{8\frac{\lambda_1(\Sigma_0)}{\log\left(1+\frac{\lambda_1(\Sigma_0)}{\sigma^2}\right)}\log(4|\mathcal{A}|/\delta)\log\left(1+\frac{\lambda_1(\Sigma_0)n}{\sigma^2}\right)}.$$

*Proof.* We note that, for each $s$, we can bound w.p. 1

$$\Gamma_{s,t} \le 4\sqrt{\frac{\lambda_1(\Sigma_0)\left(1+\frac{\lambda_1(\Sigma_q)(1+\frac{\sigma^2/\eta}{\lambda_1(\Sigma_0)})}{\lambda_1(\Sigma_0)+\sigma^2/\eta+s\lambda_1(\Sigma_q)}\right)}{\log\left(1+\frac{\lambda_1(\Sigma_0)}{\sigma^2}\left(1+\frac{\lambda_1(\Sigma_q)(1+\frac{\sigma^2/\eta}{\lambda_1(\Sigma_0)})}{\lambda_1(\Sigma_0)+\sigma^2/\eta+s\lambda_1(\Sigma_q)}\right)\right)}}\log(4|\mathcal{A}|/\delta).$$

This is true by using the upper bounds on $\sigma_{max}^2(\hat{\Sigma}_{s,t})$ in Lemma 3, and because the function $\sqrt{x/\log(1+ax)}$ for $a>0$ increases with $x$. Similarly, we have

$$\epsilon_{s,t} \le \sqrt{\delta\lambda_1(\Sigma_0)\left(1+\frac{\lambda_1(\Sigma_q)(1+\frac{\sigma^2/\eta}{\lambda_1(\Sigma_0)})}{\lambda_1(\Sigma_0)+\sigma^2/\eta+s\lambda_1(\Sigma_q)}\right)}.$$

Therefore, we have the bounds $\Gamma_{s,t} \le \Gamma_s$ w.p. 1 for all $s$ and $t$ by using appropriate $s$, and by setting $s=0$ we obtain $\Gamma$.

We are now at a position to provide the final regret bound. For any $\delta > 0$

$$R(m,n) \le \Gamma\sqrt{mnI(\mu_*;H_m)}+\mathbb{E}\sum_s\Gamma_s\sqrt{nI(\theta_{s,*};H_s\mid\mu_*,H_{s-1})}+\mathbb{E}\sum_{s,t}\epsilon_{s,t}$$

$$\le 4\underbrace{\sqrt{\frac{\lambda_1(\Sigma_q)+\lambda_1(\Sigma_0)}{\log(1+(\lambda_1(\Sigma_q)+\lambda_1(\Sigma_0))/\sigma^2)}}\log(4\mid\mathcal{A}\mid/\delta)\sqrt{mn\frac{d}{2}\log\left(1+\frac{mn\lambda_1(\Sigma_q)}{n\lambda_d(\Sigma_0)+\sigma^2}\right)}}_{\text{regret for learning }\mu}$$

$$+\sum_{s=1}^m 4\sqrt{\frac{\lambda_1(\Sigma_0)\left(1+\frac{\lambda_1(\Sigma_q)(1+\frac{\sigma^2/\eta}{\lambda_1(\Sigma_0)})}{\lambda_1(\Sigma_0)+\sigma^2/\eta+s\lambda_1(\Sigma_q)}\right)}{\log\left(1+\frac{\lambda_1(\Sigma_0)}{\sigma^2}\left(1+\frac{\lambda_1(\Sigma_q)(1+\frac{\sigma^2/\eta}{\lambda_1(\Sigma_0)})}{\lambda_1(\Sigma_0)+\sigma^2/\eta+s\lambda_1(\Sigma_q)}\right)\right)}}\log(4|\mathcal{A}|/\delta)\sqrt{n\frac{d}{2}\log\left(1+n\frac{\lambda_1(\Sigma_0)}{\sigma^2}\right)}$$

$$+\sum_{s=1}^m n\sqrt{2\delta\lambda_1(\Sigma_0)\left(1+\frac{\lambda_1(\Sigma_q)(1+\frac{\sigma^2/\eta}{\lambda_1(\Sigma_0)})}{\lambda_1(\Sigma_0)+\sigma^2/\eta+s\lambda_1(\Sigma_q)}\right)}$$

$$+2\left(\sum_{s=1}^m\sum_{t=1}^n\mathcal{E}_{s,t}\right)\mathbb{E}[\|\theta_{s,*}\|_2]$$

$$\le 4\underbrace{\sqrt{\frac{\lambda_1(\Sigma_q)+\lambda_1(\Sigma_0)}{\log(1+(\lambda_1(\Sigma_q)+\lambda_1(\Sigma_0))/\sigma^2)}}\log(4|\mathcal{A}|/\delta)\sqrt{mn\frac{d}{2}\log\left(1+\frac{mn\lambda_1(\Sigma_q)}{n\lambda_d(\Sigma_0)+\sigma^2}\right)}}_{\text{regret for learning }\mu}$$

$$+ \left( m + \frac{1}{2\lambda_1(\Sigma_0)} \sum_{s=1}^{m} \frac{\lambda_1(\Sigma_q)(\lambda_1(\Sigma_0) + \sigma^2/\eta)}{\lambda_1(\Sigma_0) + \sigma^2/\eta + s\lambda_1(\Sigma_q)} \right) \times$$

$$\left( 4\sqrt{\frac{\lambda_1(\Sigma_0)}{\log\left(1 + \frac{\lambda_1(\Sigma_0)}{\sigma^2}\right)}} \log(4|\mathcal{A}|/\delta) \sqrt{n\frac{d}{2}\log\left(1 + n\frac{\lambda_1(\Sigma_0)}{\sigma^2}\right)} + n\sqrt{2\lambda_1(\Sigma_0)\delta} \right)$$

$$+ 2md\sqrt{\|\mu_q\|_2^2 + \mathrm{tr}(\Sigma_q + \Sigma_0)}$$

$$\leq 4\underbrace{\sqrt{\frac{\lambda_1(\Sigma_q) + \lambda_1(\Sigma_0)}{\log(1 + (\lambda_1(\Sigma_q) + \lambda_1(\Sigma_0))/\sigma^2)}} \log(4|\mathcal{A}|/\delta) \sqrt{mn\frac{d}{2}\log\left(1 + \frac{mn\lambda_1(\Sigma_q)}{n\lambda_d(\Sigma_0) + \sigma^2}\right)}}_{\text{regret for learning } \mu}$$

$$+ \left( m + (1 + \frac{\sigma^2/\eta}{\lambda_1(\Sigma_0)})\log(m) \right) \times$$

$$\left( 4\sqrt{\frac{\lambda_1(\Sigma_0)}{\log\left(1 + \frac{\lambda_1(\Sigma_0)}{\sigma^2}\right)}} \log(4|\mathcal{A}|/\delta) \sqrt{n\frac{d}{2}\log\left(1 + n\frac{\lambda_1(\Sigma_0)}{\sigma^2}\right)} + n\sqrt{2\lambda_1(\Sigma_0)\delta} \right)$$

$$+ 2md\sqrt{\|\mu_q\|_2^2 + \mathrm{tr}(\Sigma_q + \Sigma_0)}$$

The first inequality follows by substituting the appropriate bounds. The second inequality first removes the part highlighted in blue (which is positive) inside the logarithm, and then uses the fact that $\sqrt{1 + x} \leq 1 + x/2$ for all $x \geq 1$. We also use $\mathbb{E}[\|\theta_{s,*}\|_2] = \sqrt{\|\mu_q\|_2^2 + \mathrm{tr}(\Sigma_q + \Sigma_0)}$ and the fact that `AdaTS` explores for $d$ rounds in each task. The final inequality replaces the summation by an integral over $s$ and derives the closed form. $\qquad\square$

# D  Proofs for Section 4.3: Semi-Bandit

In this section, we expand the linear bandit analysis to handle multiple inputs as is common in semi-bandit feedback in combinatorial optimizations. Furthermore, as the rewards for each base-arm are independent for each arm we can improve our analysis providing tighter prior dependent bounds. The center piece of the proof is again the mutual information separation between the meta-parameter and the parameter in each stage. However, in the regret decomposition we sum the confidence intervals of different arms separately.

**Notations:**  We recall the necessary notations for the proof of regret upper bound in the semi-bandit setting. For each arm $k$ the meta-parameter $\mu_{*,k} \sim \mathcal{N}(\mu_{q,k}, \sigma_{q,k}^2)$. The mean reward at the beginning for each task $s$, for an arm $k$ is sampled from $\mathcal{N}(\mu_{*,k}, \sigma_{0,k}^2)$. The reward realization of arm $k$ in round $t$ and task $s$ is denoted by $Y_{s,t}(k) = \theta_{s,*}(k) + w_{s,t}(k)$ where be the reward of the arm $k$ at time $t$ (arm $k$ need not be played during time $t$). Then the reward obtained for the action $a$ (a subset of $[K]$ with size at most $L$) is given as $Y_{s,t}(a) = \sum_{k \in a} Y_{s,t}(k)$. Let, for each task $s$ and round $t$, the action (a subset of $[K]$) be $A_{s,t}$, and the observed reward vector be $Y_{s,t} = (Y_{s,t}(k) : k \in A_{s,t})$.

The linear bandits notations for history, conditional probability, and conditional expectation carry forward to semi-bandits. Additionally, let us denote the number of pulls for arm $k$, in phase $s$, upto and excluding round $t$ as $N_{s,t}(k)$. The total number of pulls for arm $k$ in task $s$ is denoted as $N_s(k) = N_{s,n+1}(k)$, and up to and including task $s$ is denoted by $N_{1:s}(t)$.

**Mutual Information in Semi-bandits:**  The history dependent and independent mutual information terms are defined analogously, but we are now interested in the terms for each arms separately. For any arm $k \in [K]$ and action $a \subseteq [K]$, the history dependent mutual information terms of interest are

$$I_{s,t}(\theta_{s,*}(k); A_{s,t}, Y_{s,t} \mid \mu_{*,k}) = \mathbb{E}_{s,t} \left[ \frac{\mathbb{P}_{s,t}(\theta_{s,*}(k), A_{s,t}, Y_{s,t} \mid \mu_{*,k})}{\mathbb{P}_{s,t}(\theta_{s,*}(k) \mid \mu_{*,k}) \mathbb{P}_{s,t}(A_{s,t}, Y_{s,t} \mid \mu_{*,k})} \right]$$

$$I_{s,t}(\theta_{s,*}(k); a, Y_{s,t}(a) \mid \mu_{*,k}) = \mathbb{E}_{s,t} \left[ \frac{\mathbb{P}_{s,t}(\theta_{s,*}(k), Y_{s,t}(k) \mid \mu_{*,k}, A_{s,t} = a)}{\mathbb{P}_{s,t}(\theta_{s,*}(k) \mid \mu_{*,k}, A_{s,t} = a) \mathbb{P}_{s,t}(Y_{s,t}(a) \mid \mu_{*,k}, A_{s,t} = a)} \right]$$

$$I_{s,t}(\mu_{*,k}; A_{s,t}, Y_{s,t}) = \mathbb{E}_{s,t} \left[ \frac{\mathbb{P}_{s,t}(\mu_{*,k}, A_{s,t}, Y_{s,t})}{\mathbb{P}_{s,t}(\mu_{*,k}) \mathbb{P}_{s,t}(A_{s,t}, Y_{s,t})} \right]$$

$$I_{s,t}(\mu_{*,k}; a, Y_{s,t}(a)) = \mathbb{E}_{s,t} \left[ \frac{\mathbb{P}_{s,t}(\mu_{*,k}, Y_{s,t}(k) \mid A_{s,t} = a)}{\mathbb{P}_{s,t}(\theta_{s,*}(k) \mid A_{s,t}{=}a) \mathbb{P}_{s,t}(Y_{s,t}(a) \mid A_{s,t}{=}a)} \right]$$

The history mutual information independent terms of interest are

$$I(\theta_{s,*}(k); A_{s,t}, Y_{s,t} \mid \mu_{*,k}, H_{1:s,t}) = \mathbb{E}[I_{s,t}(\theta_{s,*}(k); A_{s,t}, Y_{s,t} \mid \mu_{*,k})],$$
$$I(\mu_{*,k}; A_{s,t}, Y_{s,t} \mid H_{1:s,t}) = \mathbb{E}[I_{s,t}(\mu_{*,k}; A_{s,t}, Y_{s,t})].$$

We now derive the mutual information of $\theta_{s,*}(k)$ and events in task $s$, i.e. $H_s$, given $\mu_{*,k}$, and history upto and excluding task $s$, i.e. $H_{1:s-1}$.

**Lemma 9.** *For any $k \in [K]$, $s \in [m]$, and $H_{1:s,t}$ adapted sequence of actions $((A_{s,t})_{t=1}^n)_{s=1}^m$, the following statements hold for a $(K, L)$-Semi-bandit*

$$I(\theta_{s,*}(k); H_s \mid \mu_{*,k}, H_{1:s-1}) = \mathbb{E} \sum_t \mathbb{P}_{s,t}(k \in A_{s,t}) I_{s,t}(\theta_{s,*}(k); k, Y_{s,t}(k) \mid \mu_{*,k})$$

$$I(\mu_{*,k} \mid H_{1:m}) = \mathbb{E} \sum_s \sum_t \mathbb{P}_{s,t}(k \in A_{s,t}) I_{s,t}(\mu_{*,k}; k, Y_{s,t}(k)).$$

*Proof.* The proof follows by the application of the chain rule of mutual information, and noticing that the rounds when an arm $k$ not played the mutual information $I_{s,t}(\theta_{s,*}(k); k, Y_{s,t}(k) \mid \mu_{*,k})$ and $I_{s,t}(\mu_{*,k}; k, Y_{s,t}(k))$ both are zero. This is true because no information is gained about the parameters $\theta_{s,*}(k)$ and $\mu_{*,k}$ in those rounds.

$$I(\theta_{s,*}(k); H_s \mid \mu_{*,k}, H_{1:s-1})$$

$$= \mathbb{E} \sum_t I(\theta_{s,*}(k); A_{s,t}, Y_{s,t} \mid \mu_{*,k}, H_{1:s-1}, H_{s,t-1})$$

$$= \mathbb{E} \sum_t I_{s,t}(\theta_{s,*}(k); A_{s,t}, Y_{s,t} \mid \mu_{*,k})$$

$$= \mathbb{E} \sum_t \sum_{a \in \mathcal{A}} \mathbb{P}_{s,t}(A_{s,t} = a) I_{s,t}(\theta_{s,*}(k); a, Y_{s,t}(a) \mid \mu_{*,k})$$

$$= \mathbb{E} \sum_t \sum_{a \in \mathcal{A}} \mathbb{P}_{s,t}(A_{s,t} = a) \mathbb{1}(k \in a) I_{s,t}(\theta_{s,*}(k); k, Y_{s,t}(k) \mid \mu_{*,k})$$

$$+ \mathbb{E} \sum_t \sum_{a \in \mathcal{A}} \mathbb{P}_{s,t}(A_{s,t} = a) I_{s,t}(\theta_{s,*}(k); a \setminus k, Y_{s,t}(a \setminus k) \mid \mu_{*,k}, (k, Y_{s,t}(k)))$$

$$= \mathbb{E} \sum_t \mathbb{P}_{s,t}(k \in A_{s,t}) I_{s,t}(\theta_{s,*}(k); k, Y_{s,t}(k) \mid \mu_{*,k})$$

Here, $a \setminus k$ implies the action with arm $k$ removed from subset $a$. Due to the independence of the reward of each arm, for any fixed action $a$, $\theta_{s,*}(k) \perp (a \setminus k, Y_{s,t}(a \setminus k))$ conditioned on $\mu_{*,k}$, $(k, Y_{s,t}(k))$, and history $H_{1:s,t}$. Therefore, we have

$$I_{s,t}(\theta_{s,*}(k); a \setminus k, Y_{s,t}(a \setminus k) \mid \mu_{*,k}, (k, Y_{s,t}(k))) = 0.$$

A similar sequence of steps lead to

$$I(\mu_{*,k} \mid H_{1:m}) = \mathbb{E} \sum_s \sum_t \mathbb{P}_{s,t}(k \in A_{s,t}) I_{s,t}(\mu_{*,k}; k, Y_{s,t}(k)).$$

The above equalities develop the chain rules of mutual information for each of the arms separately, by leveraging the independence of the rewards per arms. $\qquad\square$

**Per Task Regret Bound:** We derive the per task regret using the information theoretical confidence intervals while accounting for each arm separately. Let the posterior distribution of $\theta_{s,*}(k)$ at the beginning of round $t$ of task $s$ be $\mathcal{N}(\hat{\mu}_{s,t}(k), \hat{\sigma}^2_{s,t}(k))$ for appropriate $\hat{\mu}_{s,t}(k)$ and $\hat{\sigma}^2_{s,t}(k)$ that depends on the history $H_{1:s,t}$, for all $k \in [K]$, $s \in [m]$, and $t \in [n]$. We will derive these terms or bounds on these terms later.

**Lemma 10.** *For an $H_{1:s,t}$ adapted sequence of actions $((A_{s,t})_{t=1}^n)_{s=1}^m$, and any $\delta \in (0,1]$, the expected regret in round $t$ of stage $s$ in a $(K, L)$-Semi-bandit is bounded as*

$$\mathbb{E}_{s,t}[\Delta_{s,t}] = \sum_{k \in [K]} \mathbb{P}_{s,t}(k \in A_{s,t}) \left( \Gamma_{s,t}(k) \sqrt{I_{s,t}(\theta_{s,*}(k); k, Y_{s,t}(k))} + \sqrt{2\delta \tfrac{1}{K} \sigma^2_{s,t}(k)} \right), \quad (6)$$

*where*

$$\Gamma_{s,t}(k) = 4 \sqrt{ \frac{\hat{\sigma}^2_{s,t-1}(k)}{\log(1 + \hat{\sigma}^2_{s,t-1}(k)/\sigma^2)} \log(\tfrac{4K}{\delta}) }.$$

*Proof.* Similar to linear bandits we have without forced exploration

$$\mathbb{E}_{s,t}[\Delta_{s,t}] = \mathbb{E}_{s,t}[ \sum_{k \in A_{s,*}} \theta_{s,*}(k) - \sum_{k \in A_{s,t}} \theta_{s,*}(k)]$$

$$= \mathbb{E}_{s,t}[ \sum_{k \in A_{s,t}} \hat{\theta}_{s,t}(k) - \sum_{k \in A_{s,t}} \theta_{s,*}(k)]$$

$$= \mathbb{E}_{s,t}[\sum_{a \in \mathcal{A}} \mathbb{1}(A_{s,t} = a) \sum_{k \in a} (\hat{\theta}_{s,t}(k) - \theta_{s,*}(k))]$$

$$= \mathbb{E}_{s,t}[ \sum_{k \in [K]} \mathbb{1}(k \in A_{s,t})(\hat{\theta}_{s,t}(k) - \theta_{s,*}(k))]$$

$$= \sum_{k \in [K]} \mathbb{P}_{s,t}(k \in A_{s,t}) \mathbb{E}_{s,t}[\hat{\theta}_{s,t}(k) - \theta_{s,*}(k)].$$

The second equality is due to Thompson sampling ($\stackrel{d}{=}$ denotes equal distribution)

$$\sum_{k \in A_{s,*}} \theta_{s,*}(k) \mid H_{1:s,t} \stackrel{d}{=} \sum_{k \in A_{s,t}} \hat{\theta}_{s,t}(k) \mid H_{1:s,t}.$$

When forced exploration is used in some task $s$ and round $t$ we have, $\theta_{s,*}(k) \mid H_{1:s,t} \stackrel{d}{=} \hat{\theta}_{s,t}(k) \mid H_{1:s,t}$

$$
\begin{aligned}
\mathbb{E}_{s,t}[\Delta_{s,t}] &= \mathbb{E}_{s,t}[\sum_{k \in A_{s,*}} \theta_{s,*}(k) - \sum_{k \in A_{s,t}} \theta_{s,*}(k)] \\
&= \mathbb{E}_{s,t}[\sum_{k \in A_{s,t}} \hat{\theta}_{s,t}(k) - \sum_{k \in A_{s,t}} \theta_{s,*}(k)] + \mathbb{E}_{s,t}[\sum_{k \in A_{s,*}} \theta_{s,*}(k) - \sum_{k \in A_{s,t}} \hat{\theta}_{s,t}(k)] \\
&= \mathbb{E}_{s,t}[\sum_{k \in A_{s,t}} \hat{\theta}_{s,t}(k) - \sum_{k \in A_{s,t}} \theta_{s,*}(k)] + \mathbb{E}_{s,t}[\sum_{k \in A_{s,*}} \theta_{s,*}(k) - \sum_{k \in A_{s,t}} \hat{\theta}_{s,t}(k)] \\
&\leq \sum_{k \in [K]} \mathbb{P}_{s,t}(k \in A_{s,t})\mathbb{E}_{s,t}[\hat{\theta}_{s,t}(k) - \theta_{s,*}(k)] + 2\sqrt{K}\mathbb{E}_{s,t}[\sqrt{\sum_{k \in K} \theta_{s,*}^2(k)}]
\end{aligned}
$$

For each $k \in [K]$, for appropriate $\hat{\mu}_{s,t}(k)$ and $\hat{\sigma}_{s,t}^2(k)$ we know that $\hat{\theta}_{s,t}(k) \mid H_{1:s,t} \sim \mathcal{N}(\hat{\mu}_{s,t}(k), \hat{\sigma}_{s,t}^2(k))$. We define the confidence set for each arm $k$ at round $t$ of task $s$, for some $\Gamma_{s,t}(k)$, which can be a function of $H_{1:s,t}$, to be specified late, as

$$\Theta_{s,t}(k) = \{\theta :\mid \theta - \hat{\mu}_{s,t}(k) \mid \leq \tfrac{\Gamma_{s,t}(k)}{2}\sqrt{I_{s,t}(\theta_{s,*}(k); k, Y_{s,t}(k))}\}.$$

A derivation equivalent to linear bandits, gives us

$$I_{s,t}(\theta_{s,*}(k); k, Y_{s,t}(k)) = \tfrac{1}{2}\log\left(1 + \tfrac{\hat{\sigma}_{s,t-1}^2(k)}{\sigma^2}\right).$$

Because, we only consider arm $k$ we obtain as a corollary of Lemma 5 in Lu et al. [33] that for any $k$, and any $\delta\tfrac{1}{K} > 0$ for

$$\Gamma_{s,t}(k) = 4\sqrt{\frac{\hat{\sigma}_{s,t-1}^2(k)}{\log(1 + \hat{\sigma}_{s,t-1}^2(k)/\sigma^2)}\log(\tfrac{4K}{\delta})}.$$

we have $\mathbb{P}_{s,t}(\hat{\theta}_{s,t}(k) \in \Theta_{s,t}(k)) \geq 1 - \delta/2K$.

We proceed with the regret bound as

$$
\begin{aligned}
&\mathbb{E}_{s,t}[\hat{\theta}_{s,t}(k) - \theta_{s,*}(k)] \\
&\leq \mathbb{E}_{s,t}[\mathbb{1}(\hat{\theta}_{s,t}(k), \theta_{s,*}(k) \in \Theta_{s,t}(k))(\hat{\theta}_{s,t}(k) - \theta_{s,*}(k))] \\
&\quad + \mathbb{E}_{s,t}[\mathbb{1}^c(\hat{\theta}_{s,t}(k), \theta_{s,*}(k) \in \Theta_{s,t}(k))(\hat{\theta}_{s,t}(k) - \theta_{s,*}(k))] \\
&\leq \Gamma_{s,t}(k)\sqrt{I_{s,t}(\theta_{s,*}(k); k, Y_{s,t}(k))} + \sqrt{\mathbb{P}(\hat{\theta}_{s,t}(k) \text{ or } \theta_{s,*}(k) \notin \Theta_{s,t}(k))\mathbb{E}_{s,t}[(\hat{\theta}_{s,t}(k) - \theta_{s,*}(k))^2]} \\
&\leq \Gamma_{s,t}(k)\sqrt{I_{s,t}(\theta_{s,*}(k); k, Y_{s,t}(k))} + \sqrt{\delta\tfrac{1}{K}\mathbb{E}_{s,t}[(\hat{\theta}_{s,t}(k) - \hat{\mu}_{s,t}(k))^2 + (\theta_{s,*}(k) - \hat{\mu}_{s,t}(k))^2]} \\
&\leq \Gamma_{s,t}(k)\sqrt{I_{s,t}(\theta_{s,*}(k); k, Y_{s,t}(k))} + \sqrt{2\delta\tfrac{1}{K}\sigma_{s,t}^2(k)}
\end{aligned}
$$

This concludes the proof. $\qquad\square$

**Regret Decomposition:** We now develop the regret decomposition for the $(K, L)$-Semi-bandit based on the per step regret characterization in Lemma 10.

**Lemma 11.** *Let, for each $k \in [K]$, $(\Gamma_s(k))_{s \in [m]}$ and $\Gamma(k)$ be non-negative constants such that $\Gamma_{s,t}(k) \leq \Gamma_s(k) \leq \Gamma(k)$ holds for all $s \in [m]$ and $t \in [n]$ almost surely. Then for any $\delta \in (0,1]$ the*

*regret of* `AdaTS` *admits the upper bound*

$$R(m,n) \leq \sqrt{mnKL} \sqrt{\frac{1}{K} \sum_{k \in [K]} \Gamma^2(k) I(\mu_{*,k}; H_{1:m})} + 2\sqrt{mK \sum_{k \in K} \left(\mu_q^2(k) + \sigma_{0,k}^2 + \sigma_{q,k}^2\right)}$$

$$+ \sum_{s=1}^{m} \left( \sqrt{nKL} \sqrt{\frac{1}{K} \sum_{k \in [K]} \Gamma_s^2(k) I(\theta_{s,*}(k); H_s \mid \mu_{*,k}, H_{1:s-1})} + n\sqrt{2\delta \frac{1}{K} \sum_{k \in [K]} \hat{\sigma}_s^2(k)} \right).$$

*Proof.* The regret decomposition is computed in the following steps. Recall that $\mathcal{E}_{s,t}$ is the indicator if in round $t$ of task $s$ we use exploration.

$$R(m,n) = \mathbb{E}\left[\sum_{s,t} \Delta_{s,t}\right]$$

$$\leq \mathbb{E}\left[\sum_{s,t} \sum_{k \in [K]} \mathbb{P}_{s,t}(k \in A_{s,t}) \left(\Gamma_{s,t}(k) \sqrt{I_{s,t}(\theta_{s,*}(k); k, Y_{s,t}(k))}\right)\right]$$

$$+ \mathbb{E}\left[\sum_{s,t} \sum_{k \in [K]} \mathbb{P}_{s,t}(k \in A_{s,t})\epsilon_{s,t}(k)\right] + 2\mathbb{E}\left[\sum_{s,t} \mathcal{E}_{s,t}\sqrt{K}\mathbb{E}_{s,t}[\sqrt{\sum_{k \in K} \theta_{s,*}^2(k)}]\right]$$

$$\leq \mathbb{E}\left[\sum_{s,t} \sum_{k \in [K]} \mathbb{P}_{s,t}(k \in A_{s,t}) \left(\Gamma_{s,t}(k) \sqrt{I_{s,t}(\theta_{s,*}(k), \mu_{*,k}; k, Y_{s,t}(k))}\right)\right]$$

$$+ \mathbb{E}\left[\sum_{s,t} \sum_{k \in [K]} \mathbb{P}_{s,t}(k \in A_{s,t})\sqrt{2\delta \frac{1}{K} \sigma_{s,t}^2(k)}\right] + 2mK^{3/2}\mathbb{E}[\sqrt{\sum_{k \in K} \theta_{s,*}^2(k)}]$$

$$= \mathbb{E}\left[\sum_{s,t} \sum_{k \in [K]} \mathbb{P}_{s,t}(k \in A_{s,t}) \left(\Gamma_{s,t}(k) \sqrt{I_{s,t}(\theta_{s,*}(k); k, Y_{s,t}(k) \mid \mu_{*,k}) + I_{s,t}(\mu_{*,k}; k, Y_{s,t}(k))}\right)\right]$$

$$+ \mathbb{E}\left[\sum_{s,t} \sum_{k \in [K]} \mathbb{P}_{s,t}(k \in A_{s,t})\sqrt{2\delta \frac{1}{K} \sigma_{s,t}^2(k)}\right] + 2mK^{3/2} \sqrt{\sum_{k \in K} \left(\mu_q^2(k) + \sigma_{0,k}^2 + \sigma_{q,k}^2\right)}$$

$$\leq \mathbb{E}\left[\sum_{s,t} \sum_{k \in [K]} \mathbb{P}_{s,t}(k \in A_{s,t})\Gamma_{s,t}(k) \left(\sqrt{I_{s,t}(\theta_{s,*}(k); k, Y_{s,t}(k) \mid \mu_{*,k})} + \sqrt{I_{s,t}(\mu_{*,k}; k, Y_{s,t}(k))}\right)\right]$$

$$+ \mathbb{E}\left[\sum_{s,t} \sum_{k \in [K]} \mathbb{P}_{s,t}(k \in A_{s,t})\sqrt{2\delta \frac{1}{K} \sigma_{s,t}^2(k)}\right] + 2mK^{3/2} \sqrt{\sum_{k \in K} \left(\mu_q^2(k) + \sigma_{0,k}^2 + \sigma_{q,k}^2\right)}$$

$$\leq \Gamma_s(k) \sum_s \sum_{k \in [K]} \mathbb{E}\left[\sum_t \mathbb{P}_{s,t}(k \in A_{s,t})\sqrt{I_{s,t}(\theta_{s,*}(k); k, Y_{s,t}(k) \mid \mu_{*,k})}\right]$$

$$+ \sum_{k \in [K]} \Gamma(k)\mathbb{E}\left[\sum_{s,t} \mathbb{P}_{s,t}(k \in A_{s,t})\sqrt{I_{s,t}(\mu_{*,k}; k, Y_{s,t}(k))}\right]$$

$$+ \sum_s \sum_{k \in [K]} \sqrt{2\delta \frac{1}{K} \hat{\sigma}_s^2(k)}\mathbb{E}[\sum_t \mathbb{P}_{s,t}(k \in A_{s,t})] + 2mK^{3/2} \sqrt{\sum_{k \in K} \left(\mu_q^2(k) + \sigma_{0,k}^2 + \sigma_{q,k}^2\right)}$$

The first inequality follows from the expression for the reward gaps in Equation 6. The next two equations follow due to the chain rule of mutual information, similar to the linear bandit case. The only difference in this case we use the parameters for each arm ($\theta_{s,*}(k)$ and $\mu_{*,k}$) separately. Also, we use the fact that there are at most $mK$ rounds where forced exploration is used for the $m$ tasks. The next

inequality is due to $\sqrt{a+b} \leq \sqrt{a} + \sqrt{b}$. The final inequality follows as $\Gamma_{s,t}(k) \leq \Gamma_s(k) \leq \Gamma(k)$, and $\sigma_{s,t}(k) \leq \sigma_s(k)$ w.p. 1 for all $k \in [K]$, and $s \leq m$ and $t \leq n$.

We now derive the bounds for the sum of the mutual information terms for the per task parameters given the knowledge of the meta-parameter.

$$\sum_s \sum_{k \in [K]} \Gamma_s(k) \mathbb{E}\left[\sum_t \mathbb{P}_{s,t}(k \in A_{s,t}) \sqrt{I_{s,t}(\theta_{s,*}(k); k, Y_{s,t}(k) \mid \mu_{*,k})}\right]$$

$$= \sum_s \sum_{k \in [K]} \Gamma_s(k) \mathbb{E}\left[\sum_t \sqrt{\mathbb{P}_{s,t}(k \in A_{s,t})} \sqrt{\mathbb{P}_{s,t}(k \in A_{s,t}) I_{s,t}(\theta_{s,*}(k); k, Y_{s,t}(k) \mid \mu_{*,k})}\right]$$

$$\leq \sum_s \sum_{k \in [K]} \Gamma_s(k) \mathbb{E}\left[\sqrt{\sum_t \mathbb{P}_{s,t}(k \in A_{s,t})} \sqrt{\sum_t \mathbb{P}_{s,t}(k \in A_{s,t}) I_{s,t}(\theta_{s,*}(k); k, Y_{s,t}(k) \mid \mu_{*,k})}\right]$$

$$\leq \sum_s \sum_{k \in [K]} \Gamma_s(k) \sqrt{\mathbb{E} \sum_t \mathbb{P}_{s,t}(k \in A_{s,t})} \sqrt{\mathbb{E} \sum_t \mathbb{P}_{s,t}(k \in A_{s,t}) I_{s,t}(\theta_{s,*}(k); k, Y_{s,t}(k) \mid \mu_{*,k})}$$

$$= \sum_s \sum_{k \in [K]} \Gamma_s(k) \sqrt{\mathbb{E} N_s(k)} \sqrt{I(\theta_{s,*}(k); H_s \mid \mu_{*,k}, H_{1:s-1})}$$

$$\leq \sum_s \sqrt{K \sum_{k \in [K]} \mathbb{E} N_s(k)} \sqrt{\frac{1}{K} \sum_{k \in [K]} \Gamma_s^2(k) I(\theta_{s,*}(k); H_s \mid \mu_{*,k}, H_{1:s-1})}$$

$$= \sum_s \sqrt{nKL} \sqrt{\frac{1}{K} \sum_{k \in [K]} \Gamma_s^2(k) I(\theta_{s,*}(k); H_s \mid \mu_{*,k}, H_{1:s-1})}$$

The first equality is easy to see. Next sequence of inequalities follow mainly by repeated application of Cauchy-Schwarz in different forms, and application of chain rule of mutual information. We now describe the other ones.

- The second equation follow as $\sum_i a_i b_i \leq \sqrt{\sum_i a_i^2 \sum_i b_i^2}$ for $a_i, b_i \geq 0$, with $a_i = \sqrt{\mathbb{P}_{s,t}(k \in A_{s,t})}$ and $b_i = \sqrt{\mathbb{P}_{s,t}(k \in A_{s,t}) I_{s,t}(\theta_{s,*}(k); k, Y_{s,t}(k) \mid \mu_{*,k})}$.

- The third equation uses $\mathbb{E}[XY] \leq \sqrt{\mathbb{E}[X^2]\mathbb{E}[Y^2]}$ for $X, Y > 0$ w.p. 1 (positive random variables).

- The next equality first uses the relation $\mathbb{E} \sum_t \mathbb{P}_{s,t}(k \in A_{s,t}) = \mathbb{E}[N_s(k)]$ where $N_s(k)$ is the number of time arm $k$ is played in the task $s$. Then it also use the chain rule for $I(\theta_{s,*}(k); H_s \mid \mu_{*,k}, H_{1:s-1})$.

- For the next inequality, we apply Cauchy-Schwarz ($\sum_i a_i b_i \leq \sqrt{\sum_i a_i^2 \sum_i b_i^2}$) again as $a_i = \sqrt{\mathbb{E}[N_s(k)]}$ and $b_i = I(\theta_{s,*}(k); H_s \mid \mu_{*,k}, H_{1:s-1})$. Also note that $K$ and $\frac{1}{K}$ cancels out.

- The final inequality is attained by noticing $\mathbb{E}[\sum_k N_s(k)] \leq nL$, as at most $L$ arms can be played in each round.

The sum of the mutual information terms pertaining to the meta-parameter can be derived equivalently.

$$\sum_{k \in [K]} \Gamma(k) \mathbb{E}\left[\sum_{s,t} \mathbb{P}_{s,t}(k \in A_{s,t}) \sqrt{I_{s,t}(\mu_{*,k}; k, Y_{s,t}(k))}\right]$$

$$= \sum_{k \in [K]} \Gamma(k) \mathbb{E}\left[\sum_{s,t} \sqrt{\mathbb{P}_{s,t}(k \in A_{s,t})} \sqrt{\mathbb{P}_{s,t}(k \in A_{s,t}) I_{s,t}(\mu_{*,k}; k, Y_{s,t}(k))}\right]$$

$$\leq \sum_{k \in [K]} \Gamma(k) \mathbb{E}\left[\sqrt{\sum_{s,t} \mathbb{P}_{s,t}(k \in A_{s,t})} \sqrt{\sum_{s,t} \mathbb{P}_{s,t}(k \in A_{s,t}) I_{s,t}(\mu_{*,k}; k, Y_{s,t}(k))}\right]$$

$$\leq \sum_{k \in [K]} \Gamma(k) \sqrt{\mathbb{E} \sum_{s,t} \mathbb{P}_{s,t}(k \in A_{s,t})} \sqrt{\mathbb{E} \sum_{s,t} \mathbb{P}_{s,t}(k \in A_{s,t}) I_{s,t}(\mu_{*,k}; k, Y_{s,t}(k))}$$

$$= \sum_{k \in [K]} \Gamma(k) \sqrt{\mathbb{E} N_{1:m}(k)} \sqrt{I(\mu_{*,k}; H_{1:m})}$$

$$\leq \sqrt{K \sum_{k \in [K]} \mathbb{E} N_{1:m}(k)} \sqrt{\tfrac{1}{K} \sum_{k \in [K]} \Gamma^2(k) I(\mu_{*,k}; H_{1:m})}$$

$$= \sqrt{mnKL} \sqrt{\tfrac{1}{K} \sum_{k \in [K]} \Gamma^2(k) I(\mu_{*,k}; H_{1:m})}$$

For the third term we have

$$\sum_{k \in [K]} \sqrt{2\delta \tfrac{1}{K} \hat{\sigma}_s^2(k)} \mathbb{E}[\sum_t \mathbb{P}_{s,t}(k \in A_{s,t})]$$

$$\leq \sum_{k \in [K]} \sqrt{2\delta \tfrac{1}{K} \hat{\sigma}_s^2(k)} \mathbb{E}[N_s(k)]$$

$$\leq \sqrt{2\delta \tfrac{1}{K} \sum_{k \in [K]} \hat{\sigma}_s^2(k)} \sqrt{\sum_{k \in [K]} (\mathbb{E}[N_s(k)])^2}$$

$$\leq \sqrt{2\delta \tfrac{1}{K} \sum_{k \in [K]} \hat{\sigma}_s^2(k)} \sum_{k \in [K]} \mathbb{E}[N_s(k)] \leq n \sqrt{2\delta \tfrac{1}{K} \sum_{k \in [K]} \hat{\sigma}_s^2(k)}$$

This provides us with the bound stated in the lemma.

Finally, we have $\mathbb{E}[\sqrt{\sum_{k \in K} \theta_{s,*}^2(k)}] \leq \sqrt{\sum_{k \in K} \left(\mu_q^2(k) + \sigma_{0,k}^2 + \sigma_{q,k}^2\right)}$

$\square$

**Bounding Mutual Information:** The derivation of the mutual information can be done similar to the linear bandits while using the diagonal nature of the covariance matrices. We present a different argument here.

**Lemma 12.** *For any $H_{1:s,t}$-adapted action-sequence and any $s \in [m]$ and $k \in [K]$, we have*

$$I(\theta_{s,*}(k); H_s \mid \mu_{*,k}, H_{1:s-1}) \leq \tfrac{1}{2} \log \left(1 + n \tfrac{\sigma_{0,k}^2}{\sigma^2}\right) ,$$

$$I(\mu_{*,k}; H_{1:m}) \leq \tfrac{1}{2} \log \left(1 + m \tfrac{\sigma_{q,k}^2}{\sigma_{0,k}^2 + \sigma^2/n}\right) .$$

*Proof.* We have the following form for the conditional mutual information $\theta_{s,*}(k)$ with the events $H_s(k)$ conditioned on the meta-parameter $\mu_{*,k}$, and the history of arm $k$ pulls upto stage $s$ (for each $s$) $H_{1:s-1}$, as a function of $H_s$, given as

$$I(\theta_{s,*}(k); H_s \mid \mu_{*,k}, H_{1:s-1}) = \tfrac{1}{2} \mathbb{E} \log \left(1 + N_s(k) \tfrac{\sigma_{0,k}^2}{\sigma^2}\right) \leq \tfrac{1}{2} \log \left(1 + n \tfrac{\sigma_{0,k}^2}{\sigma^2}\right).$$

Another way to see this is, in each stage if $\mu_{*,k}$ was known then the variance of the estimate of $\theta_{s,*}(k)$, or equivalently of $(\theta_{s,*}(k) - \mu_{*,k})$, after $N_s(k)$ samples and with an initial variance $\sigma_{0,k}^2$ will be $\frac{1}{\sigma_0^{-2}(k) + N_s(k)\sigma^{-2}(k)}$. We note that only when $N_s(k) \geq 1$ the mutual information is non-zero. Thus we have the multiplication with $\mathbb{P}(N_s(k) \geq 1)$. The mutual information is then derived easily.

Similarly, the mutual information of $\theta_{s,*}(k)$ and the entire history of arm $k$ pulls, i.e. $H_{1:m}(k)$, is stated as follows.

$$I(\mu_{*,k}; H_{1:m}) = \tfrac{1}{2} \mathbb{E} \log \left(1 + \sum_{s=1}^{m} \frac{\sigma_{q,k}^2}{\sigma_{0,k}^2 + \sigma^2/N_s(k)}\right) \leq \tfrac{1}{2} \log \left(1 + m \frac{\sigma_{q,k}^2}{\sigma_{0,k}^2 + \sigma^2/n}\right).$$

We claim (proven shortly) that at the end of task $m$ the variance of estimate of $\mu_{*,k}$ is $(\hat{\sigma}_q^{-2}(k) + \sum_{s'=1}^{m}(\sigma_{0,k}^2 + \sigma^2/N_{s'}(k))^{-1})^{-1}$. This gives the first equality. The final inequality holds by noting that minimizing the terms $\sigma^2/N_s(k)$ with $N_s(k) = n$, for all $s$, (as any arm can be pulled at most $n$ times in any task) maximizes the mutual information.

We now derive the variance of $\mu_{*,k}$. Let the distribution of $\mu_{*,k}$ at the beginning of stage $s$ is $\mathcal{N}(\hat{\mu}_s(k), \hat{\sigma}_s^2(k))$. From the $N_s(k)$ samples of arm $k$, we know $\theta_{s,*}(k) \sim \mathcal{N}(\hat{\theta}_s(k), \sigma^2/N_s(k))$ where $\hat{\theta}_s(k)$ is the empirical mean of arm $k$ in task $s$. Further, $\theta_{s,*}(k) - \mu_{*,k} \sim \mathcal{N}(0, \sigma_{0,k}^2)$ by our reward model. Thus, we have from the two above relation

$$\mu_{*,k} \sim \mathcal{N}\left(\frac{\sigma^2/N_s(k)}{\sigma_{0,k}^2 + \sigma^2/N_s(k)}\hat{\theta}_s(k), \sigma_{0,k}^2 + \sigma^2/N_s(k)\right).$$

However, we also know independently that $\mu_{*,k} \sim \mathcal{N}(\hat{\mu}_s(k), \hat{\sigma}_s^2(k))$. Therefore, a similar combination gives us $\mu_{*,k} \sim \mathcal{N}(\hat{\mu}_{s+1}(k), \hat{\sigma}_{s+1}^2(k))$ where

$$\hat{\mu}_{s+1}(k) = \hat{\sigma}_{s+1}^{-2}(k)\left(\hat{\mu}_s(k)\hat{\sigma}_s^2(k) + \hat{\theta}_s(k)\sigma^2/N_s(k)\right)$$

$$\hat{\sigma}_{s+1}^{-2}(k) = \hat{\sigma}_s^{-2}(k) + (\sigma_{0,k}^2 + \sigma^2/N_s(k))^{-1}$$

$$\hat{\sigma}_{s+1}^{-2}(k) = \sigma_{q,k}^{-2} + \sum_{s'=1}^{s}(\sigma_{0,k}^2 + \sigma^2/N_{s'}(k))^{-1}.$$

The last equality follows from induction with the base case $\hat{\sigma}_0^2(k) = \sigma_{q,k}^2$. □

**Bounding $\Gamma_s(k)$:** We finally provide the bound on the $\Gamma_s(k)$ and $\Gamma(k)$ terms used in the regret decomposition Lemma 11.

**Lemma 13.** *For all $s \in [m]$, and $(\Gamma_s(k))_{s\in[m]}$ and $\Gamma(k)$ as defined in Lemma 11 admit the following bounds, for any $\delta \in (0,1]$, almost surely*

$$\Gamma_s(k) \le 4\sqrt{\frac{\sigma_{0,k}^2\left(1 + \frac{(1+\sigma^2/\sigma_{0,k}^2)\sigma_{q,k}^2}{(\sigma_{0,k}^2+\sigma^2)+s\sigma_{q,k}^2}\right)}{\frac{1}{2}\log\left(1 + \frac{\sigma_{0,k}^2}{\sigma^2}\left(1 + \frac{(1+\sigma^2/\sigma_{0,k}^2)\sigma_{q,k}^2}{(\sigma_{0,k}^2+\sigma^2)+s\sigma_{q,k}^2}\right)\right)}\log(\tfrac{4K}{\delta})},$$

$$\Gamma(k) \le 4\sqrt{\frac{\sigma_{0,k}^2+\sigma_{q,k}^2}{\log\left(1+\frac{\sigma_{0,k}^2+\sigma_{q,k}^2}{\sigma^2}\right)}\log(\tfrac{4K}{\delta})}.$$

*Proof.* At the beginning of task $s$ we know that $\theta_{s,*}(k) \sim \mathcal{N}(\hat{\mu}_s(k), \sigma_{0,k}^2 + \hat{\sigma}_s^2(k))$. And as the variance of $\theta_{s,*}(k)$ decreases during task $s$ with new samples from arm $k$, we have the variance

$$\hat{\sigma}_{s,t}^2(k) \le \sigma_{0,k}^2 + \hat{\sigma}_s^2(k) \le \sigma_{0,k}^2 + \frac{(\sigma_{0,k}^2+\sigma^2)\sigma_{q,k}^2}{(\sigma_{0,k}^2+\sigma^2)+s\sigma_{q,k}^2}.$$

The inequality holds by taking $N_{s'}(k) = 1$ in the expression of $\hat{\sigma}_s^2(k)$ for all tasks as arm $k$ has been played using forced exploration.

Therefore, we can bound $\Gamma_s(k)$, for any $s$, as

$$\Gamma_s(k) \le 4\sqrt{\frac{\sigma_{0,k}^2\left(1 + \frac{(1+\sigma^2/\sigma_{0,k}^2)\sigma_{q,k}^2}{(\sigma_{0,k}^2+\sigma^2)+s\sigma_{q,k}^2}\right)}{\frac{1}{2}\log\left(1 + \frac{\sigma_{0,k}^2}{\sigma^2}\left(1 + \frac{(1+\sigma^2/\sigma_{0,k}^2)\sigma_{q,k}^2}{(\sigma_{0,k}^2+\sigma^2)+s\sigma_{q,k}^2}\right)\right)}\log(\tfrac{4K}{\delta})}$$

This implies $\Gamma(k) \le 4\sqrt{\frac{\sigma_{0,k}^2+\sigma_{q,k}^2}{\log\left(1+\frac{\sigma_{0,k}^2+\sigma_{q,k}^2}{\sigma^2}\right)}\log(\tfrac{4K}{\delta})}$ by setting $s = 0$. □

**Deriving Final Regret Bound:** We proceed with our final regret bound as

**Theorem 6** (Semi-bandit). *The regret of* AdaTS *is bounded for any* $\delta \in (0,1]$ *as*

$$R(m,n) \leq \underbrace{c_1\sqrt{KLmn}}_{\text{Learning of } \mu_*} + (m+c_2)\underbrace{R_\delta(n;\mu_*)}_{\text{Per-task regret}} + \underbrace{c_3 K^{3/2}m}_{\text{Forced exploration}} + c_4\sigma\sqrt{2\delta mn},$$

*where*

$$c_1 = 4\sqrt{\frac{1}{K}\sum_{k\in[K]}\frac{\sigma_{q,k}^2+\sigma_{0,k}^2}{\log\left(1+\frac{\sigma_{q,k}^2+\sigma_{0,k}^2}{\sigma^2}\right)}\log(4K/\delta)\log\left(1+\frac{\sigma_{q,k}^2 m}{\sigma_{0,k}^2+\sigma^2/n}\right)},$$

$$c_2 = \left(1+\max_{k\in[K]:\,\sigma_{0,k}>0}\frac{\sigma^2}{\sigma_{0,k}^2}\right)\log m, \quad c_3 = 2\sqrt{\sum_{k\in[K]}(\mu_{q,k}^2+\sigma_{q,k}^2+\sigma_{0,k}^2)},$$

$$c_4 = \sqrt{\frac{1}{K}\sum_{k\in[K]:\,\sigma_{0,k}=0}\log\left(1+\frac{\sigma_{q,k}^2 m}{\sigma^2}\right)}.$$

*The* per-task regret *is bounded as* $R_\delta(n;\mu_*) \leq c_5\sqrt{KLn}+\sqrt{2\delta\frac{1}{K}\sum_{k\in[K]}\sigma_{0,k}^2 n}$*, where*

$$c_5 = 4\sqrt{\frac{1}{K}\sum_{k\in[K]:\,\sigma_{0,k}>0}\frac{\sigma_{0,k}^2}{\log\left(1+\frac{\sigma_{0,k}^2}{\sigma^2}\right)}\log(4K/\delta)\log\left(1+\frac{\sigma_{0,k}^2 n}{\sigma^2}\right)}.$$

*The prior widths* $\sigma_{q,k}$ *and* $\sigma_{0,k}$ *are defined as in Section 3.1.*

*Proof.* We now use the regret decomposition in Lemma 11, the bounds on terms $\Gamma_s(k)$ and $\Gamma(k)$ in Lemma 13, and the mutual information in Lemma 12 bounds derived earlier to obtain our final regret bound for the semi-bandits. The regret bound follows from the following chain of inequalities.

$R(m,n)$

$$\leq \sqrt{mnKL}\sqrt{\frac{1}{K}\sum_{k\in[K]}\Gamma^2(k)I(\mu_{*,k};H_{1:m})}+2\sqrt{mK\sum_{k\in K}\left(\mu_q^2(k)+\sigma_{0,k}^2+\sigma_{q,k}^2\right)}$$

$$+\sum_s\left(\sqrt{nKL}\sqrt{\frac{1}{K}\sum_{k\in[K]}\Gamma_s^2(k)I(\theta_{s,*}(k);H_s\mid\mu_{*,k},H_{1:s-1})}+n\sqrt{2\delta\frac{1}{K}\sum_{k\in[K]}\hat\sigma_s^2(k)}\right)$$

$$\leq 4\sqrt{\frac{1}{K}\sum_{k\in[K]}\frac{\sigma_{0,k}^2+\sigma_{q,k}^2}{\log\left(1+\frac{\sigma_{0,k}^2+\sigma_{q,k}^2)}{\sigma^2}\right)}\log\left(1+m\frac{\sigma_{q,k}^2}{\sigma_{0,k}^2+\sigma^2/n}\right)\log(\tfrac{4K}{\delta})}\sqrt{mnKL}$$

$$+2mK^{3/2}\sqrt{\sum_{k\in K}\left(\mu_q^2(k)+\sigma_{0,k}^2+\sigma_{q,k}^2\right)}$$

$$+\sum_s\left(\sqrt{nKL}\sqrt{\frac{1}{K}\sum_{k\in[K]:\sigma_{0,k}^2=0}\Gamma_s^2(k)I(\theta_{s,*}(k);H_s\mid\mu_{*,k},H_{1:s-1})}+n\sqrt{2\delta\frac{1}{K}\sum_{k\in[K]:\sigma_{0,k}^2=0}\hat\sigma_s^2(k)}\right)$$

$$+\sum_s\left(\sqrt{nKL}\sqrt{\frac{1}{K}\sum_{k\in[K]:\sigma_{0,k}^2>0}\Gamma_s^2(k)I(\theta_{s,*}(k);H_s\mid\mu_{*,k},H_{1:s-1})}+n\sqrt{2\delta\frac{1}{K}\sum_{k\in[K]:\sigma_{0,k}^2>0}\hat\sigma_s^2(k)}\right)$$

$$\leq 4\sqrt{\frac{1}{K}\sum_{k\in[K]}\frac{\sigma_{0,k}^2+\sigma_{q,k}^2}{\log\left(1+\frac{\sigma_{0,k}^2+\sigma_{q,k}^2)}{\sigma^2}\right)}\log\left(1+m\frac{\sigma_{q,k}^2}{\sigma_{0,k}^2+\sigma^2/n}\right)\log(\tfrac{4K}{\delta})}\sqrt{mnKL}$$

$$+2mK^{3/2}\sqrt{\sum_{k\in K}\left(\mu_q^2(k)+\sigma_{0,k}^2+\sigma_{q,k}^2\right)}+n\sqrt{2m\delta\frac{1}{K}\sum_{k\in[K]:\sigma_{0,k}^2=0}\sum_{s=1}^m\frac{\sigma^2\sigma_{q,k}^2}{\sigma^2+s\sigma_{q,k}^2}}$$

$$+ \left( m + \sum_{s=1}^{m} \max_{k\in[K]:\sigma^2_{0,k}>0} \frac{1}{2\sigma^2_{0,k}} \frac{(\sigma^2_{0,k}+\sigma^2)\sigma^2_{q,k}}{(\sigma^2_{0,k}+\sigma^2)+s\sigma^2_{q,k}} \right)$$

$$\times \left( 4\sqrt{\frac{1}{K} \sum_{k\in[K]:\sigma^2_{0,k}>0} \frac{\sigma^2_{0,k}}{\log\left(1+\frac{\sigma^2_{0,k}}{\sigma^2}\right)} \log\left(1+n\frac{\sigma^2_{0,k}}{\sigma^2}\right)} \log(\tfrac{4K}{\delta})\sqrt{nKL} + n\sqrt{2\delta\frac{1}{K}\sum_{k\in[K]:\sigma^2_{0,k}>0}\sigma^2_{0,k}} \right)$$

$$\leq 4\sqrt{\frac{1}{K}\sum_{k\in[K]} \frac{\sigma^2_{0,k}+\sigma^2_{q,k}}{\log\left(1+\frac{\sigma^2_{0,k}+\sigma^2_{q,k})}{\sigma^2}\right)} \log\left(1+m\frac{\sigma^2_{q,k}}{\sigma^2_{0,k}+\sigma^2/n}\right)} \log(\tfrac{4K}{\delta})\sqrt{mnKL}$$

$$+ 2mK^{3/2}\sqrt{\sum_{k\in K}\left(\mu_q^2(k)+\sigma^2_{0,k}+\sigma^2_{q,k}\right)} + n\sqrt{2\delta\sigma^2 m\frac{1}{K}\sum_{k\in[K]:\sigma^2_{0,k}=0}\log(1+m\frac{\sigma^2_{q,k}}{\sigma^2})}$$

$$+ \left( m + \left(1 + \max_{k\in[K]:\sigma^2_{0,k}>0} \frac{\sigma^2}{\sigma^2_{0,k}}\right)\log(m) \right)$$

$$\times \left( 4\sqrt{\frac{1}{K} \sum_{k\in[K]:\sigma^2_{0,k}>0} \frac{\sigma^2_{0,k}}{\log\left(1+\frac{\sigma^2_{0,k}}{\sigma^2}\right)} \log\left(1+n\frac{\sigma^2_{0,k}}{\sigma^2}\right)} \log(\tfrac{4K}{\delta})\sqrt{nKL} + n\sqrt{2\delta\frac{1}{K}\sum_{k\in[K]:\sigma^2_{0,k}>0}\sigma^2_{0,k}} \right)$$

The derivation follows through steps similar to the corresponding derivations for the linear bandits. In the second inequality we differentiate the arms which has $\sigma^2_{0,k} = 0$ against the rest. Any arm $k$ with $\sigma^2_{0,k} = 0$ has no mutual information once $\mu_{*,k}$ is known, i.e. $I(\theta_{s,*}(k); H_s \mid \mu_{*,k}, H_{1:s-1}) = 0$ for all such $k$. □

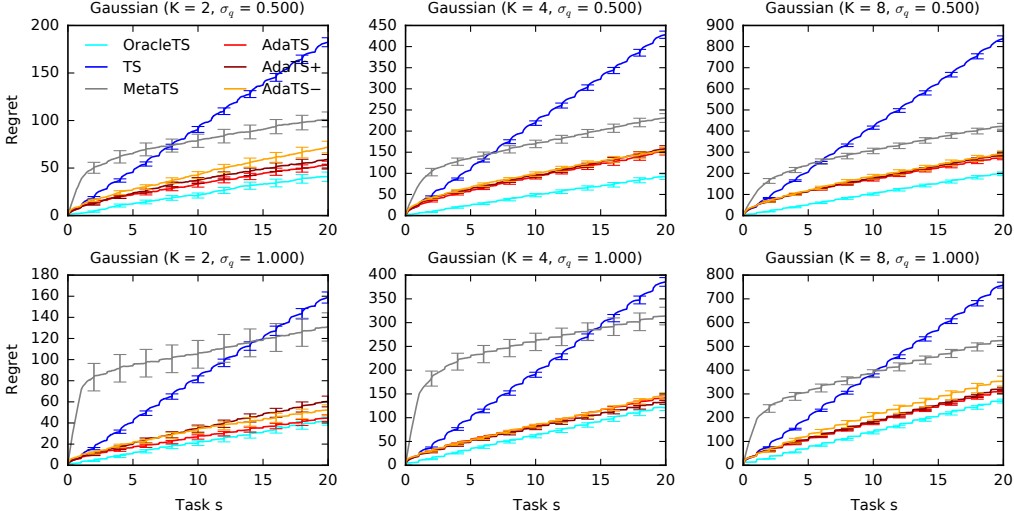

Figure 4: `AdaTS` in a $K$-armed Gaussian bandit. We vary both $K$ and meta-prior width $\sigma_q$.

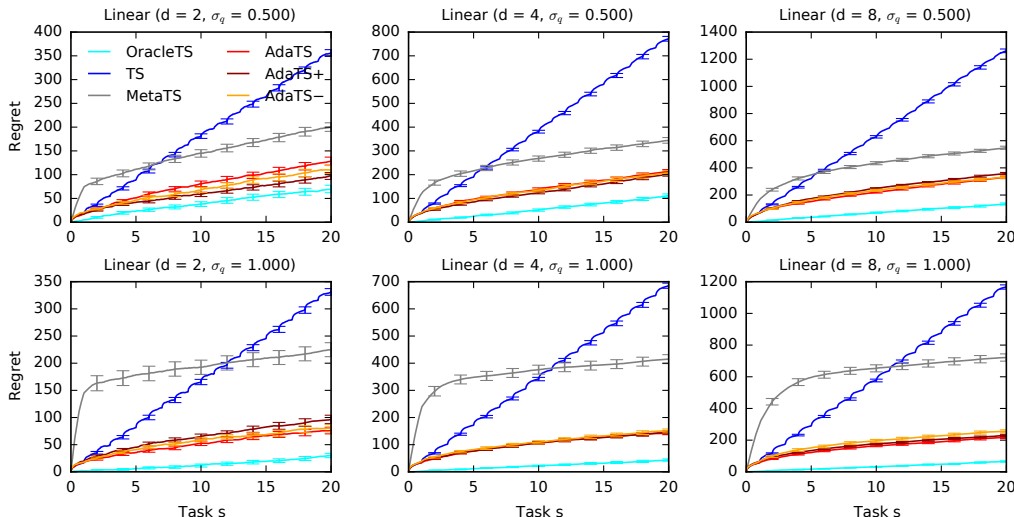

Figure 5: `AdaTS` in a $d$-dimensional linear bandit with $K = 5d$ arms. We vary both $d$ and meta-prior width $\sigma_q$.

## E    Supplementary Experiments

We conduct two additional experiments. In Appendix E.1, we extend synthetic experiments from Section 5. In Appendix E.2, we experiment with two real-world classification problems: MNIST [32] and Omniglot [30].

### E.1    Synthetic Experiments

This section extends experiments in Section 5 in three aspects. First, we show the Gaussian bandit with $K \in \{2, 4, 8\}$ arms. Second, we show the linear bandit with $d \in \{2, 4, 8\}$ dimensions. Third, we implement `AdaTS` with a misspecified meta-prior.

Our results are reported in Figures 4 and 5. The setup of this experiment is the same as in Figure 2, and it confirms all earlier findings. We also experiment with two variants of misspecified `AdaTS`. In `AdaTS`$^+$, the meta-prior width is widened to $3\sigma_q$. This represents an overoptimistic agent. In `AdaTS`$^-$, the meta-prior width is reduced to $\sigma_q/3$. This represents a conservative agent. We observe that this misspecification has no major impact on the regret of `AdaTS`, which attests to its robustness.

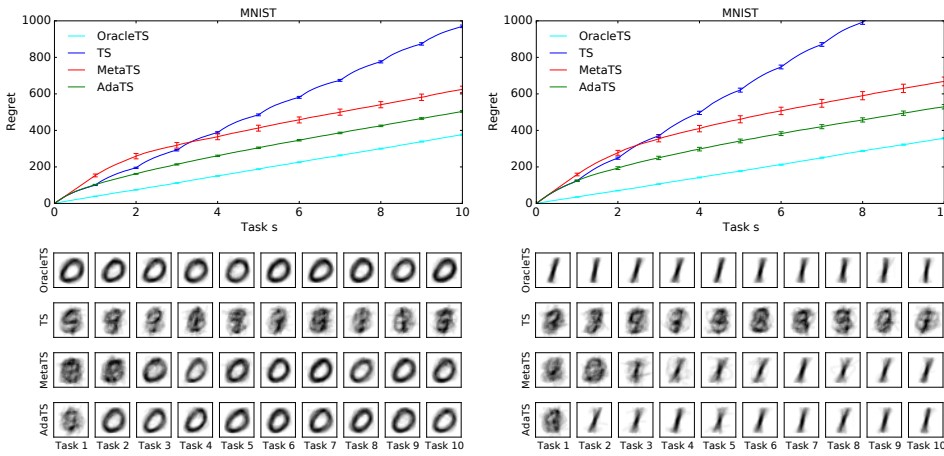

Figure 6: `AdaTS` in two meta-learning problems of digit classification from MNIST. On the top, we plot the cumulative regret as it accumulates over rounds within each task. Below we visualize the average digit, corresponding to the pulled arms in round 1 of the tasks.

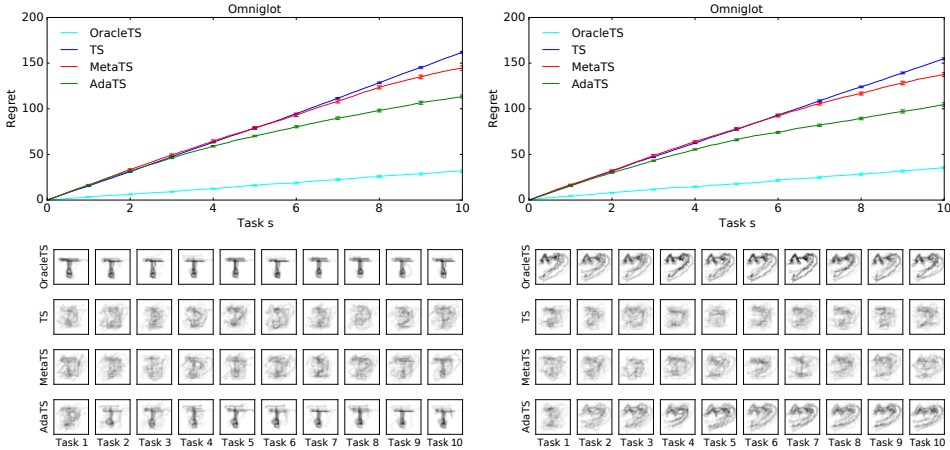

Figure 7: `AdaTS` in two meta-learning problems of character classification from Omniglot. On the top, we plot the cumulative regret as it accumulates over rounds within each task. Below we visualize the average character, corresponding to the pulled arms in round 1 of the tasks.

### E.2 Online One-Versus-All Classification Experiments

We consider online classification on two real-world datasets, which are commonly used in meta-learning. The problem is cast as a multi-task linear bandit with Bernoulli rewards. Specifically, we have a sequence of image classification tasks where one class is selected randomly to be positive. In each task, at every round, $K$ random images are selected as the arms and the goal is to pull the arm corresponding to an image from the positive class. The reward of an image from the positive class is $\mathrm{Ber}(0.9)$ and for all other classes is $\mathrm{Ber}(0.1)$. Dataset-specific settings are as follows:

1. **MNIST** [32]: The dataset contains $60\,000$ images of handwritten digits, which we split into equal-size training and test sets. We down-sample each image to $d = 49$ features and then use these as arm features. The training set is used to estimate $\mu_0$ and $\Sigma_0$ for each digit. The bandit algorithms are evaluated on the test set. In each simulation, we have $m = 10$ tasks with horizon $n = 200$ and $K = 30$ arms.

2. **Omniglot** [30]: The dataset contains $1\,623$ different handwritten characters from $50$ different alphabets. This is an extremely challenging dataset because we have only $20$ human-drawn images per character. Therefore, it is important to adapt quickly. We train a $4$-layer CNN to extract $d = 64$ features using characters from $30$ alphabets. The remaining $20$ alphabets are

split into equal-size training and test sets, with 10 images per character in each. The training set is used to estimate $\mu_0$ and $\Sigma_0$ for each character. The bandit algorithms are evaluated on the test set. In each simulation, we have $m = 10$ tasks with horizon $n = 10$ and $K = 10$ arms. We guarantee that at least one character from the positive class is among the $K$ arms.

In all problems, the meta-prior is $\mathcal{N}(\mathbf{0}, I_d)$ and the reward noise is $\sigma = 0.1$. We compare AdaTS with the same three baselines as in Section 5, repeat all experiments 20 times, and report the results in Figures 6 and 7. Along with the cumulative regret, we also visualize the average digit / character corresponding to the pulled arms in round 1 of each task. We observe that AdaTS learns a very good meta-parameter $\mu_*$ almost instantly, since its average digit / character in task 2 already resembles the unknown highly-rewarding digit / character. This happens even in Omniglot, where the horizon of each task is only $n = 10$ rounds. Note that the meta-prior was not selected in any dataset-specific way. The fact that AdaTS still works well attests to the robustness of our method.