# OpenReview forum: "No Regrets for Learning the Prior in Bandits"
_NeurIPS.cc/2021/Conference — NeurIPS 2021 Poster_

### Official Review · Reviewer_7ezS · 2021-07-15

**Rating:** 6
**Confidence:** 3

**Summary:**

This paper studies a variant of Thompson Sampling for meta-learning. By maintaining the distribution of the parameter of the prior distribution, the proposed algorithm can adapt to the certain task it encountered with. The authors also provide Bayesian regret bound for this algorithm. Experiments are provided to support their theoretical claim showing their algorithm can outperform benchmark algorithms.

**Limitations And Societal Impact:**

No potential social impact is found. Regarding the presentation for the proof techniques, I suggest the authors to highlight the connection between theirs and MetaTS's. For example, can the MetaTS proof technique can be used in this proof. If not, what's the major difficulty here?

**Main Review:**

This paper is overall clear and easy to follow and the authors also provide some examples showing the proposed general algorithm can be adapted to several bandit settings. One minor concern I have is regarding the novelty of this paper. This work could be considered as a follow-up work of MetaTS[1], where MetaTS samples $\mu_s$ and here the authors directly use the distribution of $\mu_s$ then sample the rewards. These two sampling mechanisms are similar theoretically. Although the author claimed that the proposed sampling method would be more robust, in general cases, will it be more difficult to implement compared to MetaTS since it needs to handle the distribution over $\mu_s$ explicitly?

Another question is, since the MetaTS does not provide the proof for linear gaussian bandits, I wonder if the proof technique of MetaTS can be applied to this work?

Due to the contribution of this paper, I'd like to suggest marginally accept this paper.

[1] Kveton, Branislav, et al. "Meta-Thompson Sampling." International Conference on Machine Learning (ICML), 2021.

**Time Spent Reviewing:**

4

---

> ### Author Response · Authors · 2021-08-09
> **Rebuttal**
>
> Thank you for the detailed review. We clarify the difference between MetaTS and AdaTS below.
>
> **Algorithmic difference between MetaTS and AdaTS**
>
> Sampling in MetaTS is not theoretically identical to AdaTS. MetaTS samples the meta-parameter $\mu_s$ at the beginning of each task $s$ and uses it to compute the posterior of the task parameter $\theta_{s, \ast}$. Since $\mu_s$ is fixed within the task, MetaTS does not have a correct posterior of $\theta_{s, \ast}$ given the past. AdaTS marginalizes out the uncertainty in the meta-parameter $\mu_\ast$, and thus has a correct posterior of $\theta_{s, \ast}$ within the task. We discuss this in lines 108-113. This seemingly minor difference leads to superior empirical results (Figure 2 in the main paper and Figures 4-7 in Appendix E) and a fully-Bayesian analysis. A fully-Bayesian analysis is not possible for MetaTS.
>
> **Extending MetaTS analysis to linear bandits**
>
> We believe that the analysis can be extended. However, the price of meta-learning would be $O(\sqrt{m} n^2)$, owing to the worst-case nature of the analysis of MetaTS. This penalty arises due sampling the meta-parameter $\mu_s$ at the beginning of each task $s$. The price of meta-learning in our work is $O(\sqrt{m n})$, a huge improvement.
>
> **Generality of AdaTS**
>
> We are not aware of any problem instance where we could apply MetaTS but not AdaTS. Thus, to the best of our knowledge, AdaTS is at least as general as MetaTS.

---

> > ### Comment · Reviewer_7ezS · 2021-08-26
> > **Thank you for your response**
> >
> > Thank the authors for their response, I have carefully read their response and choose to stick to my score

---

### Official Review · Reviewer_x5jf · 2021-07-16

**Rating:** 7
**Confidence:** 4

**Summary:**

This work presents a Thompson sampling algorithm for the Bayesian "meta-learning" bandit problem, where bandit instances are drawn from a "task" distribution of known form with unknown location parameter, itself drawn from a known meta-prior.

The key ideas in the proposed AdaTS algorithm are
1. to maintain a distribution over the unknown task prior parameter, and
2. to marginalize such uncertainties by learning, across instances, a posterior over it.

Leveraging hierarchical Bayesian modeling and conjugate priors, the authors present AdaTS, with implementations for multi-armed/linear bandits and combinatorial semi-bandits.

Bayesian regret bounds for the $n$-round $m$-instance AdaTS are shown to be of order $O(\sqrt{mn})$.

Experiments on synthetic datasets illustrate the added benefits of the proposed approach.

**Limitations And Societal Impact:**

The authors discuss the limitations of a frequentist Vs Bayesian regret analysis point of view.

The authors should more clearly convey the necessary inputs to the proposed algorithm:
- the form and variance parameters of the task distribution $P(\cdot)$ ---besides the form and parameters of the meta-prior $Q$ already specified.

**Main Review:**

The authors present a hierarchical view of the "meta-learning" bandit problem and leverage the Bayesian modeling toolkit and Thompson sampling to propose AdaTS.

The work is very clearly presented, with a concise introduction (Section 1), problem setting (Section 2) and algorithm (Section 3) presentations.

The novelty of Proposition 1 (and its proof) is unclear, as it looks like a direct (sequential) application of Bayes rule. Similarly, even though they state that equations in Sections 3.2 and 3.3 are due to Kveton et al. [27], these appear to be a direct application of the conjugate prior based analysis of the Gaussian likelihood (e.g., https://en.wikipedia.org/wiki/Conjugate_prior#When_likelihood_function_is_a_continuous_distribution).
- I would encourage the authors to explain/justify whether these are direct results of Gaussian conjugate prior based Bayesian posterior updates, or if there is any technical novelty on the presented content.

AdaTS is proposed in Algorithm 1, whose originality lies on realizing that the parameter sampling step introduced by MetaTS [27] introduces high variance, which can be avoided via marginalization. In other words, the authors leverage Rao-Blackwellization (well known in the estimation theory community) to improve upon an existing algorithm that leads to tighter regret bounds and better empirical performance.

Section 4 contains the theoretical analysis of the proposed algorithm, which is an information-theoretic technique that adapts the analysis of [32] to derive and proof Lemma 2: a general bound on AdaTS that conveys the price of not knowing the task prior parameter $\mu^*$ and the regret when $\mu^*$ is known.

- A minor comment on the proof presentation of Section 4 is that in equation (3) the expectation has subscripts $s$ and $t$, to probably indicate the probability distribution over which this expectation is taken, but has not been previously defined (this is the only case where expectation is denoted with subscripts).

The authors then proceed to bound the key regret terms for the linear bandit (Section 4.2) and Gaussian semi-bandit cases (Section 4.3). The authors provide an explanation of the proof approach, and discuss the impact of priors (as a function of both the meta-prior and the task-prior parameters), showing that the price of not knowing the task-prior parameter is "small".

The experiments section showcases the benefits of the proposed approach, as
1. the hierarchical modeling allows for learning across instances, and
2. marginalization improves regret.

- Since all the studied algorithms are Thompson sampling based, a suggestion here would be to consider some of the UCB-based metalearning algorithms discussed in Section 6.

A final comment relates to the fact that AdaTS requires knowledge of the reward variance parameters $\Sigma_0$ (which should be explicitly indicated in Algorithm 1):
- An open question (somehow related to the use of Bayesian posterior updates) is whether AdaTS could be extended to the unknown $\Sigma_0$ variance case (as this would imply t-distributed task distributions).

**Time Spent Reviewing:**

2

---

> ### Author Response · Authors · 2021-08-09
> **Rebuttal**
>
> Thank you for the detailed review. Our response is below.
>
> **Novelty in Proposition 1**
>
> In the bandit setting, actions are chosen adaptively based on the history. Therefore, we carefully rederive the sequential Bayes rule to account for this adaptation. The adaptation does not cause any issues and we get the expected result, as you pointed out.
>
> **Conjugate priors in Section 3.2 and Section 3.3**
>
> The wiki articles that you pointed out are insufficient to derive our updates, because they consider only the standard notion of conjugancy. Our model is hierarchical (Figure 1). Specifically, in the notation of Figure 1, the wiki articles can be used to derive a posterior of $\theta_{s, \ast}$ given observations $Y_{s, t}$, for any task $s$. However, in our derivations, we have a posterior of the meta-parameter $\mu_\ast$ given observations $Y_{\ell, t}$ in all prior tasks $\ell < s$, where we need to marginalize $\theta_{\ell, \ast}$ out.
>
> **Unknown $\Sigma_0$**
>
> For a single task with parameter $\theta \sim N(\mu_\ast, \Sigma_0)$, and unknown $\mu_\ast$ and $\Sigma_0$, a normal-inverse-Wishart conjugate prior exits. However, in our setting, we would need a conjugate prior for the normal-inverse-Wishart prior. We are not aware of what this prior would be and leave this for future work.
>
> **Other minor issues**
>
> We will clarify the subscripts $s$ and $t$. We will also clearly state that the form and variance of $P$ are known, and that $Q$ is known exactly.

---

### Official Review · Reviewer_PsGp · 2021-07-17

**Rating:** 6
**Confidence:** 4

**Summary:**

The paper considers the meta Thompson Sampling problem with hierarchical structure. In particular, paper considers a setting where there are m-bandits instances and within each instance, there are n-time periods as well K-arms. Here, within an instance, the parameters of the distribution corresponding to arms are sampled from a common distribution, whose parameters themselves follow a prior distribution (hence the name meta). For this problem, the paper presents a Thompson Sampling algorithm, AdaTS and proves Bayesian Regret bounds under a few common prior distributions. The regret bounds of \sqrt{mn} seems to improve over the existing state of the art regret bounds.

**Limitations And Societal Impact:**

Theory paper, no implications to society

**Main Review:**

The main contribution I take away from the paper is that the paper presents a more efficient sampling algorithm that achieves improved regret bounds of O(\sqrt{mn}) in comparison to existing bounds from O(n^2 \sqrt{m}). While this seems to be an interesting and relevant contribution, I believe the technicality contributions of this paper are incremental, which I will elaborate below. Furthermore, the paper does not provide motivating application for this problem set up. Without the motivating application, it was extremely hard to understand the problem set up and how different layers interact with each other. Connection to a realistic application, would have helped the reader appreciate the improvements in regret bounds more. I also strongly believe there is a lot of room for improving the technical presentation.

1. The issue of unknown \mu* in the analysis. On page 2, the paper seems to indicate that if \mu* is known, the existing technical machinery in Russo and Van Roy can be leveraged to obtain the regret of O(m\sqrt{Kn}). It appears that the paper attempts to match this bound without knowing \mu*. Recently, there is a growing literature on the min-max regret of Thompson Sampling algorithm for a wide range of reward distribution settings including the ones considered in the paper ( see [A],  [B], [C],  [D]). Given that min-max regret is a much stronger regret notion than the Bayesian Regret (i.e. Bayesian Regret would trivially follow from min-max regret) and the existing approaches do not require the knowledge of prior, why can't these approaches be trivially extended to the obtain the regret of O(m\sqrt{Kn})? Does this explain why it is not surprising that we will have no regret even when \mu* is unknown?

2. Ignoring the hierarchical structure of the problem. In the introduction, the paper provides a vague example of how ignoring the structure could be sub-optimal under certain settings. However, in the context of above example, it seems like the goal is to obtain a regret bound of O(m\sqrt{Kn}), isn't that inconsistent with the expectation that this is sub-optimal? The paper also does not specifically provide bounds for the standard bandit setting (with Bernoulli rewards) which made it harder for me to reconcile with the above inconsistency. I guess one can generalize the results for Linear bandit to the standard bandit setting with K-arms by considering a K-dimensional feature vector. However, the bounds in Theorem-5 does not seem to indicate any dependency on the dimension of the feature vector or number of arms, what am I missing here?

3. Interpreting Theorem 5: Line 255 on page 7 mentions that mR_\delta is bounded as O(m\sqrt{n}). However, by line 258, the paper claims the bound is only O(\sqrt{mn}). What am I missing?




[A] S. Agrawal, N. Goyal, "Further optimal regret bounds for Thompson Sampling", In Proceedings of the 16th International Conference on Artificial Intelligence and Statistics (AISTATS), 2013

[B] S. Agrawal, N. Goyal, "Thompson Sampling for contextual bandits with linear payoffs". In Proceedings of the 30th International Conference on Machine Learning (ICML), 2013.

[C] S. Wang, W. Chen. Thompson Sampling for Combinatorial Semi-Bandits. In Proceedings of the 35th International Conference on Machine Learning, 2018.

[D] K. Jun, A. Bhargava, R. Nowak, R. Willett. Scalable Generalized Linear Bandits: Online Computation and Hashing. Part of Advances in Neural Information Processing Systems 30 (NIPS 2017).

**Time Spent Reviewing:**

5 hrs

---

> ### Author Response · Authors · 2021-08-09
> **Rebuttal**
>
> Thank you for the detailed review. Our response is below. In all claims, we omit all logarithmic factors but $\log K$, to make the dependence on the number of arms $K$ clear.
>
> **Motivating example**
>
> We present a general framework for learning to explore from similar past exploration problems. Potential applications are:
>
> 1. Cold-start personalization in recommender systems where users are tasks. The users have similar preferences, but neither the individual preferences nor their similarity is known in advance.
>
> 2. Online regression with bandit feedback (Figure 3 and Appendix E) where individual regression problems are tasks. Similar examples in the tasks have similar mean responses, which are unknown in advance.
>
> 3. Industrial applications, such as VLSI testing, where a series of silicon wafers is tested. Testing each chip in a wafer is expensive, but the failures usually occur close to each other in a group. The goal is to quickly find regions with failures without testing all chips in the wafer. In our terminology, each task corresponds to a wafer and arm pulls correspond to testing different dies in that wafer. Since the wafers are manufactured using a similar process, they have similar failure regions, but these regions are unknown in advance.
>
> **Bayes versus min-max regret**
>
> The Bayes regret can characterize the hardness of a problem instance by how informative the prior is. The min-max regret considers the worst case and cannot do that. Specifically, consider a Bayesian linear bandit with $K$ arms where the parameter $\theta \in R^d$ is sampled from $N(\mu_\ast, \Sigma_0)$. The $n$-round min-max regret of TS is $O(d \sqrt{n \log K})$. However, the Bayes regret scales as $O(\sqrt{\lambda_1(\Sigma_0) d n \log K})$, where $\lambda_1(M)$ is the maximum eigenvalue of $M$. The Bayes regret naturally captures the fact that the regret decreases as the prior becomes more informative, to zero as $\lambda_1(\Sigma_0) \to 0$. Therefore, it is a perfect metric for learning to explore better, and we use it in our work. In our setting, we learn $\mu_\ast$ above.
>
> **Regret improvement due to using structure**
>
> In our linear bandit (Sections 3.2 and 4.2), the meta-parameter is sampled as $\mu_\ast \sim N(\mu_q, \Sigma_q)$ and we have $m$ exploration tasks. The parameter in task $s$ is generated as $\theta_{s, \ast} \sim N(\mu_\ast, \Sigma_0)$. If $\mu_\ast$ was known, TS with prior $N(\mu_\ast, \Sigma_0)$ would have a Bayes regret of
>
> $$O(m \sqrt{\lambda_1(\Sigma_0) d n \log K}).$$
>
> This is a theoretical optimum (cyan lines in Figure 2). When the hierarchy of the parameter sampling process is ignored, each task parameter can be viewed as $\theta_{s, \ast} \sim N(\mu_\ast, \Sigma_q + \Sigma_0)$, which leads to a Bayes regret of
>
> $$O(m \sqrt{\lambda_1(\Sigma_q + \Sigma_0) d n \log K}).$$
>
> Since $\Sigma_q$ and $\Sigma_0$ are PSD, $\lambda_1(\Sigma_q + \Sigma_0) \geq \lambda_1(\Sigma_0)$, and this is more costly (blue lines in Figure 2) than if $\mu_\ast$ was known. Finally, AdaTS (red lines in Figure 2) uses the hierarchy of our problem to learn $\mu_\ast$ over time and has a Bayes regret of
>
> $$O(m \sqrt{\lambda_1(\Sigma_0) d n \log K} +
> \sqrt{\lambda_1(\Sigma_q + \Sigma_0) d m n \log K}).$$
>
> This is what Theorem 5 states, and is a huge improvement when $m$ is large and $\lambda_1(\Sigma_q + \Sigma_0) \gg \lambda_1(\Sigma_0)$.
>
> **Theorem 5**
>
> The dependence on the number of arms $|\mathcal{A}|$ and dimension $d$ can be seen between lines 249 and 251. To instantiate Theorem 5 in a $K$-armed Gaussian bandit, set $d = |\mathcal{A}| = K$. The result would be a Bayes regret bound of
>
> $$O(m \sqrt{\lambda_1(\Sigma_0) K n \log K} +
> \sqrt{\lambda_1(\Sigma_q + \Sigma_0) K m n \log K}).$$
>
> The extra $\sqrt{\log K}$ can be eliminated by instantiating Theorem 6 with $L = 1$.
>
> We apologize for the confusion in lines 255 and 258. Line 255 is the regret of solving $m$ tasks under the assumption that $\mu_\ast$ is known, which is
>
> $$O(m \sqrt{\lambda_1(\Sigma_0) d n \log K}).$$
>
> Line 258 is the additional regret due to meta-learning $\mu_\ast$, which is
>
> $$O(\sqrt{\lambda_1(\Sigma_q + \Sigma_0) d m n \log K}).$$
>
> **Bernoulli bandit**
>
> We do not analyze Bernoulli bandits, since this would require analyzing a different meta-parameter model (Section 3.4). However, note that Gaussian bandits are as standard as Bernoulli bandits.

---

### Author Response · Authors · 2021-08-19
**Discussion with authors**

Dear reviewers,

We believe that we properly and factually addressed all major points raised in your reviews. Please let us know if you have any other concerns. We would be happy to discuss them.

Sincerely,

The authors

---

### Decision · Program_Chairs · 2021-09-27

**Decision:**

Accept (Poster)

**Comment:**

This paper studies a meta learning for bandits problem, using a hierarchical Bayesian formulation. It proposes the AdaTS algorithm, which can be instantiated to Gaussian {multi-armed, linear, combinatorial semi} bandits, and more general exponential family bandits.

Specialized to the Gaussian bandits setting, the proposed AdaTS algorithm improves over two baselines, including (1) the prior work of MetaTS, in its regret in learning the task prior parameter \mu_* from \sqrt{mn^2} to \sqrt{mn} (2) Thompson sampling without meta-learning, in that when the task prior is sufficiently concentrated and the meta prior is sufficiently "spread out". Empirical results clearly support the claims. One limitation of the current work is that it does not have theoretical results yet in the more general exponential family bandits setting (e.g. Bernoulli bandits).